# Hydrolysis-deficient mosaic microtubules as faithful mimics of the GTP cap

Juan Estévez-Gallego [1,2] ✉, Thorsten B. Blum [2,6], Felix Ruhnow [1,6], María Gili[1], Silvia Speroni[1], Raquel García-Castellanos[1], Michel O. Steinmetz [2,3] & Thomas Surrey [1,4,5] ✉

A critical feature of microtubules is their GTP cap, a stabilizing GTP-tubulin rich region at growing microtubule ends. Microtubules polymerized in the presence of GTP analogs or from GTP hydrolysis-deficient tubulin mutants have been used as GTP-cap mimics for structural and biochemical studies. However, these analogs and mutants generate microtubules with diverse biochemical properties and lattice structures, leaving it unclear what is the most faithful GTP mimic and hence the structure of the GTP cap. Here, we generate a hydrolysis-deficient human tubulin mutant, αE254Q, with the smallest possible modification. We show that αE254Q-microtubules are stable, but still exhibit mild mutation-induced growth abnormalities. However, mixing two GTP hydrolysis-deficient tubulin mutants, αE254Q and αE254N, at an optimized ratio eliminates growth and lattice abnormalities, indicating that these 'mosaic microtubules' are faithful GTP cap mimics. Their cryo-electron microscopy structure reveals that longitudinal lattice expansion, but not protofilament twist, is the primary structural feature distinguishing the GTP-tubulin containing cap from the GDP-tubulin containing microtubule shaft. However, alterations in protofilament twist may be transiently needed to allow lattice compaction and GTP hydrolysis. Together, our results provide insights into the structural origin of GTP cap stability, the pathway of GTP hydrolysis and hence microtubule dynamic instability.

Microtubules are tubular filaments composed of αβ-tubulin heterodimers, which adopt a pseudo-helical geometry[1]. They are essential for various cellular functions, such as intracellular transport during interphase and chromosome segregation during cell division[2,3]. Microtubules switch stochastically between periods of growth and shrinkage, a property called dynamic instability, which is critical for their function[4,5]. Microtubules are stabilized at their growing ends by the so-called "GTP cap", a region enriched in GTP-tubulin as a result of delayed nucleotide hydrolysis after GTP-tubulin addition to microtubule ends (Fig. 1A)[6,7]. The transition from growth to shrinkage, called

catastrophe, is thought to be caused by the stochastic loss of the GTP cap exposing the unstable GDP-tubulin-containing microtubule shaft at microtubule ends. Hence, to understand the fundamental mechanism of dynamic instability it is critical to understand the structural properties of the GTP cap[8].

The structure of the microtubule lattice in the GTP-cap region differs from the one of the GDP-microtubule, allowing members of the end binding (EB) protein family to selectively recognize the GTP cap, binding there with about ten times higher affinity than to the GDP-tubulin-containing microtubule shaft[9]. Fluorescently labeled EBs form

[1]Centre for Genomic Regulation (CRG), The Barcelona Institute of Science and Technology, Barcelona, Spain. [2]Laboratory of Biomolecular Research, Division of Biology and Chemistry, Paul Scherrer Institut, Villigen, Switzerland. [3]University of Basel, Biozentrum, Basel, Switzerland. [4]Universitat Pompeu Fabra (UPF), Barcelona, Spain. [5]ICREA, Pg. Lluis Companys 23, Barcelona, Spain. [6]These authors contributed equally: Thorsten B. Blum, Felix Ruhnow.
✉e-mail: juan.estevez-gallego@psi.ch; thomas.surrey@crg.eu

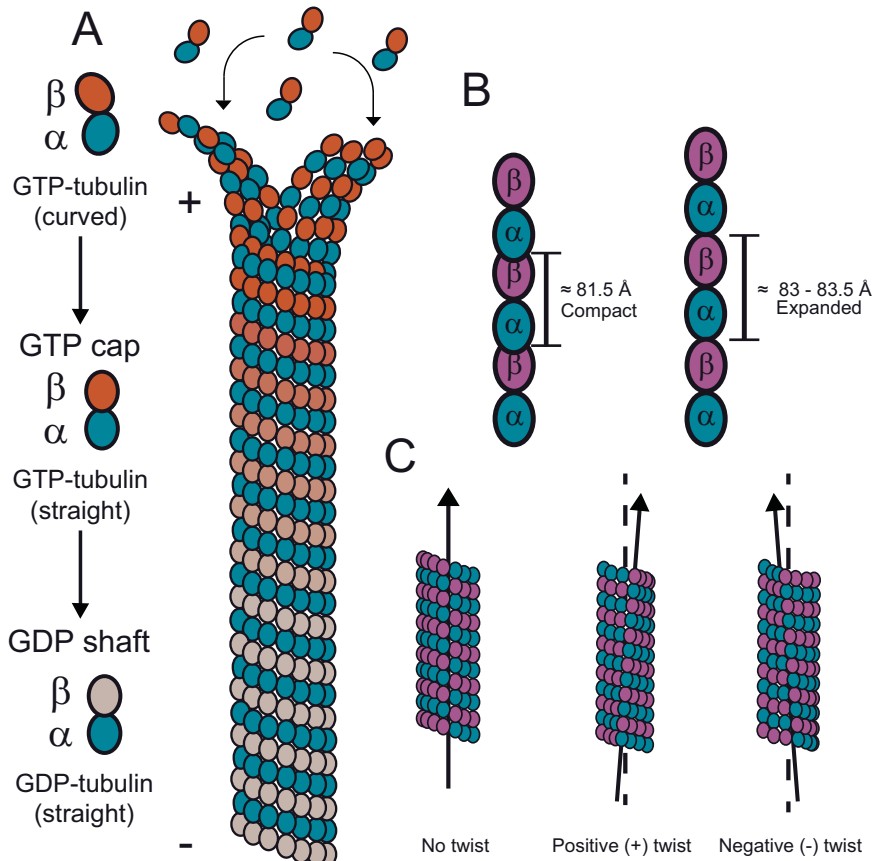

**Fig. 1 | Schematic of microtubule structural properties. A** Schematic representation of a growing microtubule. Curved GTP-tubulin heterodimers are incorporated at the microtubule plus (+) end, forming the GTP cap (indicated by orange β-tubulins). GTP-tubulin is gradually lost towards the minus (−) end due to GTP hydrolysis, causing the microtubule shaft to contain GDP-tubulin (indicated by gray β-tubulins). **B**, **C** Schematic representations illustrating the two main microtubule lattice parameters, dimer rise (**B**) and protofilament twist (**C**).

characteristic "comets" at growing microtubule ends, both in cells[10–12] and when reconstituted in vitro[13–15]. These comets form because the density of high-affinity binding sites decreases over a distance of hundreds of nanometers from the growing microtubule end due to GTP hydrolysis[15,16]. EBs can therefore label the transiently existing GTP-cap region at the growing ends of microtubules in fluorescence microscopy experiments.

Cryo-electron microscopy (cryo-EM) has been successfully used to determine the structure of microtubules up to ~3 Å resolution[17–20]. This has clearly established the structure of the GDP-microtubule lattice, but a high-resolution structure of the GTP cap at growing microtubule ends is still lacking. This is mainly due to the heterogeneity of growing ends displaying "tapered" and/or "flared" protofilaments with varying lengths and curvatures[1,21,22] and the stochastically fluctuating size of the GTP-tubulin-containing microtubule end region[23,24]. These characteristics prevent successful averaging of microtubule cap micrographs, which is, however, required to obtain high-resolution reconstructions from cryo-EM data.

Instead, non-hydrolysable GTP analogs were used to polymerize static mimics of GTP-tubulin-containing microtubules (denoted GTP-microtubules from here onwards) that could be analyzed by cryo-EM. However, contrasting results obtained with different nucleotide analogs raised the question of which analog is a good GTP mimic. GMPCPP is one of the most prominent GTP analogs, forming microtubules that suppress dynamic instability, as expected for a GTP-microtubule mimic[25]. GMPCPP-microtubules exhibit a slight increase in the average tubulin dimer rise by 1.5–2.0 Å compared to mammalian GDP microtubules, generating an axial lattice expansion (Fig. 1B) accompanied by a slight right-handed or positive protofilament twist (Fig. 1C)[26,27]. These lattice conformational characteristics were interpreted as exerting a stabilizing effect on the GMPCPP microtubule structure[19,20,26]. However, GMPCPP-microtubules show relatively weak EB binding, raising the question to which extent GMPCPP-microtubules represent a faithful mimic of a GTP cap[9,28]. On the other hand, other potential GTP analogs such as GTPγS or GDP-BeF$_3^-$ produced microtubules to which EBs bound with high affinity[9], but these showed a compacted lattice (Fig. 1B) and a left-handed or negative protofilament twist (Fig. 1C), different from GMPCPP-microtubules[29,30].

To overcome the use of GTP analogs, hydrolysis-deficient recombinant tubulin mutants were recently produced to polymerize microtubules that contain indeed GTP-tubulin throughout their entire lattice instead of nucleotide analog-tubulin. GTP hydrolysis was blocked by mutating the catalytic glutamate residue at position 254 of α-tubulin (αE254) to alanine (αE254A)[31] or to asparagine (αE254N) in human tubulin[32]. Microtubules polymerized from either mutant tubulin were stable, showing no catastrophe events as expected for a GTP-microtubule. Both mutant microtubules displayed an expanded lattice (like GMPCPP-microtubules) but had very different protofilament twists and different preferences for protofilament number and helical pitch of the tubulins in the lattice[32]. These structural differences are accompanied by biochemical differences: whereas αE254A-microtubules nucleated extremely efficiently and bound EBs with high affinity all along the lattice[31], αE254N-microtubules did not nucleate well and surprisingly displayed segments with differing EB-binding affinities[32] (Table 1).

**Table 1 | Summary of biochemical properties of tubulin variants**

|  | WT | αE254A | αE254N | αE254Q |
|---|---|---|---|---|
| Nucleation | + | +++ | + | ++ |
| Growth speed | ++ | ? | + | +++ |
| Stability | +/– | + | + | + |
| Curvature | + | ? | + | ++ |
| EB binding | Growing ends | Homogeneous | Heterogeneous | Homogeneous |
| EB affinity | ++ at ends | +++ | n/d | +++ |
| Reference | 31 | 31 | 32 | This work |

Interestingly, these observations show that GTP analogs and single point mutations blocking GTP hydrolysis in human tubulin all produce stable microtubule lattices, despite being biochemically and structurally diverse. This reveals a remarkable sensitivity of the microtubule lattice to small changes in the catalytic site, either induced by mutation or nucleotide modification. Therefore, the question arises whether a good mimic of a wild-type GTP-microtubule lattice exists at all, without introducing any nucleotide analog or mutation-specific effects beyond blocking GTP hydrolysis.

Here, we generated a hydrolysis-deficient human tubulin mutant with the smallest possible modification by replacing the catalytic glutamate residue by a glutamine (αE254Q). Microtubules polymerized from this tubulin mutant contain GTP, are stable, and bind EB3 uniformly with high affinity. However, αE254Q-microtubules still display some mutation-specific growth abnormalities, even though to a much milder extent compared to microtubules grown from previously produced tubulin mutants. Moreover, we could demonstrate that polymerizing 'mosaic microtubules' consisting of a mixture of two mutants, αE254Q, and αE254N-tubulin at an optimized ratio, can eliminate mutation-specific biochemical effects, thus generating a faithful mimic of the GTP cap. Cryo-EM analysis of these mosaic GTP-microtubules revealed that GTP hydrolysis is accompanied by compaction of an initially expanded lattice, but no major change in protofilament twist. However, the transient adoption of a negative protofilament twist appears to be required to facilitate compaction. Our results isolate the major structural and biochemical lattice characteristics associated with stable GTP-microtubules, providing insights into the structural properties of the GTP cap and the pathway of GTP hydrolysis in the GTP cap.

## Results

### Human αE254Q-tubulin forms stable GTP-microtubules
To generate a GTPase-deficient mutant with the smallest possible modification compared to wild-type tubulin (WT-tubulin) we replaced the catalytic glutamate residue 254 of α-tubulin by glutamine. Human αβ-tubulin containing the αE254Q mutation (αE254Q-tubulin) was expressed in insect cells and purified as previously described[31] with some modifications (see "Methods") (Fig. 2A). The nucleotide content of microtubules polymerized from αE254Q-tubulin in the presence of GTP was analyzed by reverse-phase ion-pair HPLC chromatography, confirming that in contrast to WT, human αE254Q-microtubules are indeed hydrolysis-deficient and contain GTP throughout their lattice, like the previously generated αE254A-microtubules (Fig. 2B, C).

Next, we examined the dynamic properties of individual microtubules polymerized from αE254Q-tubulin using simultaneous interference reflection microscopy (IRM) and total internal reflection fluorescence (TIRF) microscopy[33,34]. Microtubules were polymerized from surface-immobilized CF640R-labeled GMPCPP-stabilized microtubule seeds in the presence of unlabeled αE254Q-tubulin, mGFP-labeled EB3 and GTP (see Methods). EB3 bound all along the stable αE254Q-microtubules (Fig. 2D), which polymerized without displaying any catastrophes (Fig. 2E). This was in contrast to dynamically growing and shrinking WT microtubules displaying EB3 comets only at their growing ends (Fig. 2D, E and Supplementary Fig. 1), as expected[31]. Hence, αE254Q-microtubules show characteristics that agree with having GTP bound throughout their entire lattice instead of only at their growing ends.

### αE254Q-microtubules display GTP-lattice and mutation-specific properties
The high-affinity binding of EB3 all along αE254Q-microtubules is similar to αE254A-microtubules[31], but distinct from αE254N-microtubules that display segmented EB3 binding to low and high-affinity microtubule segments, indicative of a certain uncoupling of the nucleotide and lattice conformational state in some parts of the microtubule[32]. EB3 bound to αE254Q-microtubules with a dissociation constant of $8.3 \pm 2.9$ nM, as determined from measuring the mGFP-EB3 fluorescence intensity along the length of αE254Q-microtubules that were pre-polymerized from GMPCPP seeds, surface-immobilized, and then incubated with varying concentrations of mGFP-EB3 (Fig. 2F, G). This dissociation constant is similar to that measured previously for GTP hydrolysis-deficient αE254A-microtubules ($7.8 \pm 1.1$ nM) (Table 2)[31]. Further, αE254Q-tubulin nucleated relatively mildly, while αE254A-tubulin was previously observed to nucleate extremely efficiently and αE254N-tubulin to nucleate poorly. Hence, the nucleation and growth properties of αE254Q-tubulin deviate less from the wild-type behavior compared to other mutants (Table 1). However, αE254Q-microtubules appeared unnaturally curved when co-polymerized with EB3, in contrast to the notably straighter WT microtubules (Fig. 2D and Supplementary Fig. 1A).

Together, these observations indicate that although all described αE254-tubulin mutants generate a GTP-microtubule lattice because of inhibited GTP hydrolysis, they also display distinct, mutation-specific effects on some biochemical and morphological microtubule properties. These findings indicate that even the smallest modification of tubulin at its critical catalytic residue αE254 affects the microtubule lattice properties, possibly due to the highly repetitive nature of the microtubule lattice amplifying small changes when repeated throughout. This raises the question of how the effect of the presence of GTP-tubulin in the microtubule lattice can be isolated from the effects that the mutations themselves can have.

### Mixing mutant tubulins in mosaic microtubules eliminates mutation-specific effects
In order to try to reduce or even eliminate the mutation-specific effects in GTP hydrolysis-deficient microtubules, we polymerized microtubules from mixtures of two different tubulin mutants, generating "mosaic microtubules". We reasoned that randomizing the positioning of different mutations in the microtubule could prevent the amplification of the same local structural change by the repetitive and potentially cooperative nature of the microtubule lattice. To this end, we chose the αE254Q- and αE254N-tubulin mutants as they harbor the smallest changes as compared to WT-tubulin.

Starting from pure αE254N-tubulin, we increased the percentage of αE254Q-tubulin in the mixture used for microtubule polymerization, keeping the total tubulin concentration at 10 µM. Using TIRF

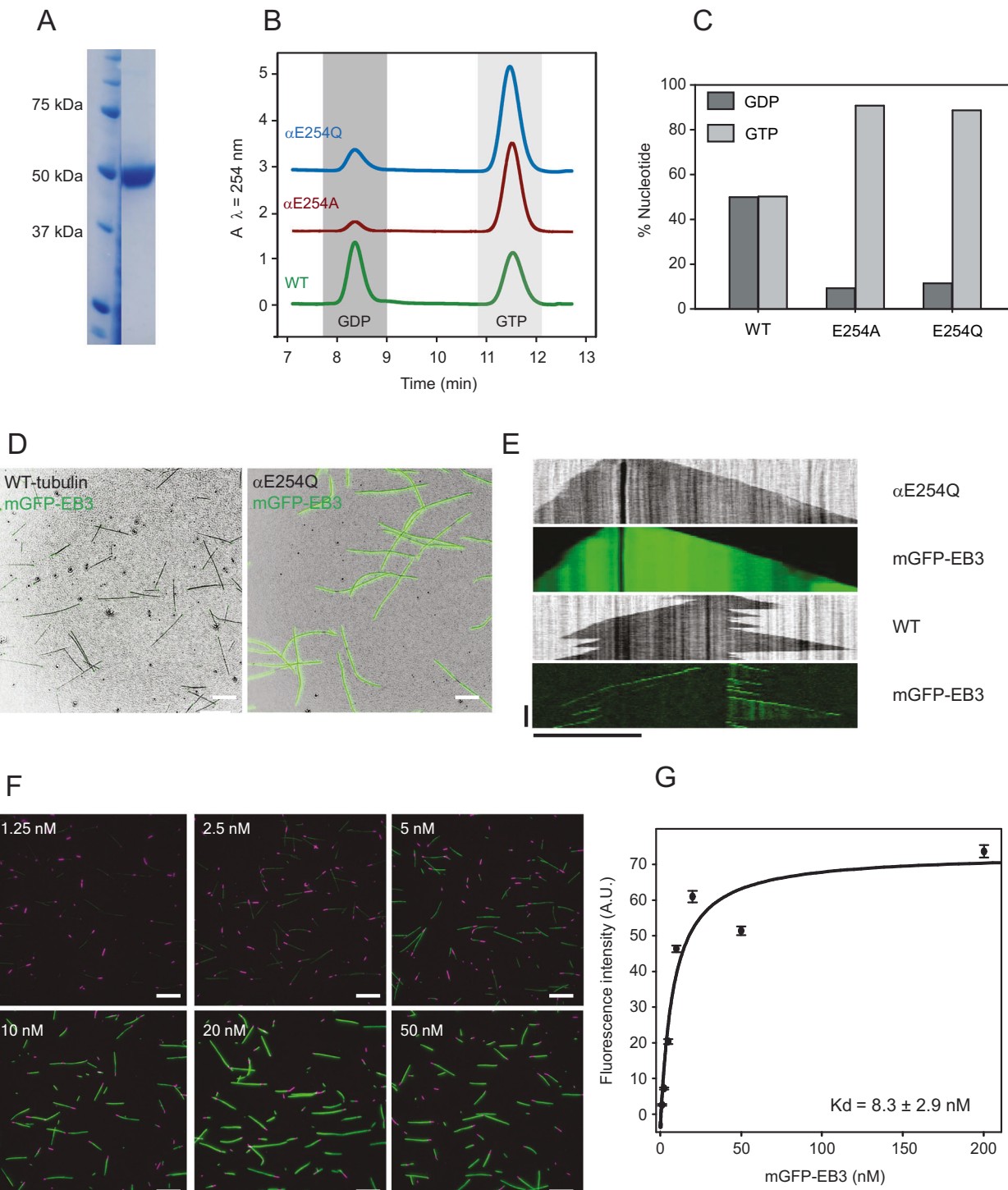

**Fig. 2 | Characterization of αE254Q-microtubules. A** Coomassie-stained SDS gel of purified recombinant human αE254Q-tubulin (right lane). The quality of the purified protein in the gel is representative of three independent purifications ($N = 3$) from separate batches of αE254Q-tubulin-expressing insect cells (for original purification gels see the Source Data file). **B** Example HPLC chromatograms of nucleotides extracted from microtubules composed of WT (green), αE254A (red), and αE254Q (blue) recombinant tubulin. Dark and light gray boxes highlight the areas of the GDP peak (elution time ≈ 8.4 min) and the GTP peak (elution time = 11.6 min), respectively. **C** Bar graph showing the mean nucleotide composition of WT, αE254A, and αE254Q-microtubules. Dark and light gray bars represent the average percentages of GDP and GTP, respectively. $N = 2$ independent experiments; $n = 2$ replicas for each condition. Error bars represent SEM. **D** Overlaid IRM and TIRF microscopy images of αE254Q-microtubules (black) polymerized for 20 min from glass-attached GMPCPP microtubule seeds (black) in the presence of 40 nM mGFP-EB3 (green). $N = 4$ independent experiments. **E** Example IRM/TIRF microscopy kymographs of WT and αE254Q-microtubules. **F** TIRF microscopy images of αE254Q-microtubules in the presence of different concentrations of mGFP-EB3 (green). αE254Q-microtubules were pre-grown in solution from CF640R-labeled "GMPCPP-seeds" (purple) and then surface-immobilized, followed by the addition of mGFP-EB3. **G** Plot of average maximum mGFP-EB3 fluorescence intensities measured along αE254Q-microtubules at the different EB3 concentrations indicated in (**F**). A hyperbolic fit (black line) yields the dissociation constant, $K_d$, as indicated. $N = 2$ independent experiments; $n = 100$ microtubules per each mGFP-EB3 concentration. Error bars represent the SEM. Fluorescence intensities are represented in arbitrary units (A.U.). White and black horizontal scale bars represent 10 µm; vertical scale bar represents 5 min. Source data are provided as a Source Data file.

**Table 2 | Calculated biochemical parameters ± SEM**

|  | WT | αE254Q | αE254N | 20%Q + N |
|---|---|---|---|---|
| $k_{on,(app)}$ ($\mu M^{-1} s^{-1}$) | 3.9 ± 0.3 | 6.2 ± 0.3 | 2.6 ± 0.5 | 3.7 ± 0.1 |
| $k_{off,(app)}$ ($s^{-1}$) | 7.0 ± 4.3 | 3.4 ± 1.6 | 11.9 ± 7.5 | 0.9 ± 1.5 |
| $L_{p(app)}$ ($\mu m$) | 416 ± 88 | 240 ± 38 | – | 520 ± 31 |
| $K^*$ ($\mu M^{-1} s^{-1} 10^{-5}$) | 60.0 ± 7.6 | 170.0 ± 5.8 | – | 71.7 ± 5.9 |
| mGFP-EB3 Kd (nM) | 39.6 ± 5.8[*] | 8.3 ± 2.9 | – | 14.0 ± 2.6 |

[*]Kd for comets[31].

microscopy in the presence of mGFP-EB3, we observed that at all tubulin mutant ratios microtubules grew and were stable (Fig. 3A). Remarkably, already in the presence of 5–10% of αE254Q-tubulin in the mixture, we observed that the peculiar segmented mGFP-EB3 binding that is characteristic for pure αE254N-microtubules disappeared, which we demonstrated quantitatively by linescan analysis of mGFP-EB3 intensities along αE254Q/αE254N mosaic microtubules (Supplementary Fig. 2). Hence, even a low percentage of αE254Q-tubulin is sufficient to abolish the mutation-specific effect of αE254N-tubulin on lattice segmentation. Increasing the percentage of αE254Q-tubulin in mosaic microtubules beyond 30% caused increasingly curved and discontinuous microtubule growth, the characteristic properties observed for pure αE254Q-microtubules (Fig. 3A, inset kymographs). Hence, a relatively high percentage of αE254N-tubulin in mosaic microtubules appears to be needed to abolish the mutation-specific effect of αE254Q-tubulin on microtubule curvature.

Next, we used the plus end growth speeds of mosaic microtubules as an additional biochemical criterion for determining their optimal composition for a good GTP cap mimic. Increasing the percentage of αE254Q-tubulin in the tubulin mixture gradually increased the growth speed of mosaic microtubules formed in the presence of 10 μM total tubulin in a non-linear manner (Fig. 3B). At 20% αE254Q- and 80% αE254N-tubulin (referred to as 20%Q + N-tubulin), mosaic microtubules grew steadily with the speed of recombinant WT microtubules which grew at ~21 nm/s (Supplementary Fig. 3 and Supplementary Table 1). This was remarkable, because at around the same composition of mutant tubulins, mutation-specific effects on other microtubule properties such as segmented EB3 binding and microtubule curvature were also eliminated (Fig. 3A, C).

To test whether the different tubulin mutants incorporate in the mosaic microtubules at the same 20:80 ratio as present in solution, we conducted mass spectrometry experiments and found that 20%Q + N-microtubules indeed consisted of 18.9 ± 4.3% of αE254Q-tubulin and the rest being αE254N-tubulin (Supplementary Fig. 4). Hence, 20%Q + N-microtubules add tubulins at a similar rate as the wild-type GTP cap, generating a GTP-microtubule lattice without noticeable tubulin mutation-specific properties other than displaying blocked GTP hydrolysis.

## Mosaic 20%Q + N-microtubules represent a faithful GTP cap mimic

Next, we quantified several other biochemical properties of 20%Q + N-microtubules in order to compare them to the properties of the GTP cap at the growing ends of WT microtubules. First, using TIRF microscopy, we performed titration experiments of 20%Q + N-microtubules to measure how much mGFP-EB3 bound to 20%Q + N-microtubules. mGFP-EB3 revealed high affinity with a dissociation constant of 14.0 ± 2.6 nM (Fig. 4A, Table 2, and Supplementary Fig. 5A), similar to that observed for pure αE254Q-microtubules (Fig. 2G). This affinity is also close to the affinity of EB3 binding to the wild-type GTP cap (39.6 ± 5.8 nM)[16,31], in agreement with these mosaic microtubules being a faithful GTP-cap mimic.

Using TIRF microscopy, we measured the dependence of the plus end growth speed of 20%Q + N-microtubules on the tubulin concentration and compared it to the respective dependencies of pure

αE254Q-, αE254N- and WT microtubules (Fig. 4B, C and Supplementary Table 1). All growth speeds increased linearly with the tubulin concentration, as expected[35–37]. Remarkably, 20%Q + N-microtubules grew with speeds characteristically different from αE254Q- and αE254N-microtubules and very similar to WT microtubules at all tubulin concentrations. Extracting the apparent on-rate constant ($k_{on,(app)}$, Table 2) from the slope of a linear fit to the data revealed very similar values for 20%Q + N (3.7 ± 0.1 $\mu M^{-1} s^{-1}$) and WT microtubules (3.9 ± 0.3 $\mu M^{-1} s^{-1}$), values that are characteristically different from those for αE254Q and αE254N-microtubules (6.2 ± 0.3 $\mu M^{-1} s^{-1}$ and 2.6 ± 0.5 $\mu M^{-1} s^{-1}$, respectively). These results demonstrate that the kinetic growth properties of the plus ends of 20%Q + N- and WT microtubules are very similar (Table 2). Such a close similarity over such an extended concentration range has to our knowledge not been reported for other GTP cap mimics, including GMPCPP-microtubules[35,38].

Next, we compared how efficiently 20%Q + N-microtubules nucleated in solution, expecting that a good mimic of the GTP cap would exhibit spontaneous microtubule nucleation comparable to that of wild-type GTP-tubulin. In turbidity experiments, we measured the time course of the absorbance at 350 nm of microtubule suspensions growing at different tubulin concentrations (Fig. 4D and Supplementary Fig. 5B). The inverse time required to reach 10% of the absorbance maximum was plotted against the tubulin concentration (Fig. 4D, Inset)[39]. The slopes of the linear regression (K*) representing the apparent nucleation rates (Fig. 4E and Table 2) confirmed that αE254Q-tubulin exhibits robust, spontaneous nucleation (170 ± 5.8 $\mu M^{-1} s^{-1} 10^{-5}$). Contrary, WT-tubulin (60 ± 7.6 $\mu M^{-1} s^{-1} 10^{-5}$) and the 20%Q + N mixture (71.7 ± 5.9 $\mu M^{-1} s^{-1} 10^{-5}$) displayed modest and very similar spontaneous microtubule nucleation, as indicated by their almost parallel linear regressions. Inspection of these samples by IRM confirmed that the measured increase of turbidity even at high tubulin concentrations (20 μM) is due to an increase of the mass of WT-, αE254Q-, and 20%Q + N-microtubules (Fig. 4F). In these IRM experiments, we could observe the formation of αE254A-microtubules already at ten times lower concentrations (2 μM), as previously described[31] (Fig. 4F, bottom right). Therefore, the 20%Q + N-tubulin mixture produces GTP-microtubules at a rate closest to wild-type GTP-tubulin, in contrast to other GTP cap mimics that either nucleate much more or less efficiently.

Finally, we also quantified the curvature of WT-, αE254Q-, and 20% Q + N-microtubules by determining their persistence length. Microtubules were elongated from immobilized "seeds" at similar growth speeds by adjusting the respective tubulin concentrations based on the known concentration dependencies of the growth speed (Fig. 4B), and their persistence length was estimated using curvature distribution analysis (Supplementary Fig. 5C, D)[40]. In contrast to the curved αE254Q-microtubules with an average persistence length of only 240 ± 38 μm, the average persistence length of 20%Q + N-microtubules was 520 ± 31 μm, more similar to the 416 ± 88 μm of recombinant WT microtubules (Table 2). This value is a little higher than that measured previously for bovine brain microtubules using the same method (310 ± 90 μm)[40], potentially indicating certain species or isotype differences. This analysis confirms that the mutation-specific effect of αE254Q-tubulin on microtubule curvature can indeed be avoided by mixing it with αE254N-tubulin in 20%Q + N-microtubules.

Together, these results further support the notion that tubulin mutation-specific effects on microtubule morphology and biochemical properties, going beyond blocking GTP hydrolysis, have been eliminated in the 20%Q + N-microtubules and that they can be therefore considered as a faithful mimic of the GTP-cap.

## Lattice parameters of αE254Q and 20%Q + N-microtubules

Next, we used cryo-EM to determine the structural lattice characteristics of αE254Q and 20%Q + N-microtubules[41]. To this end, we imaged

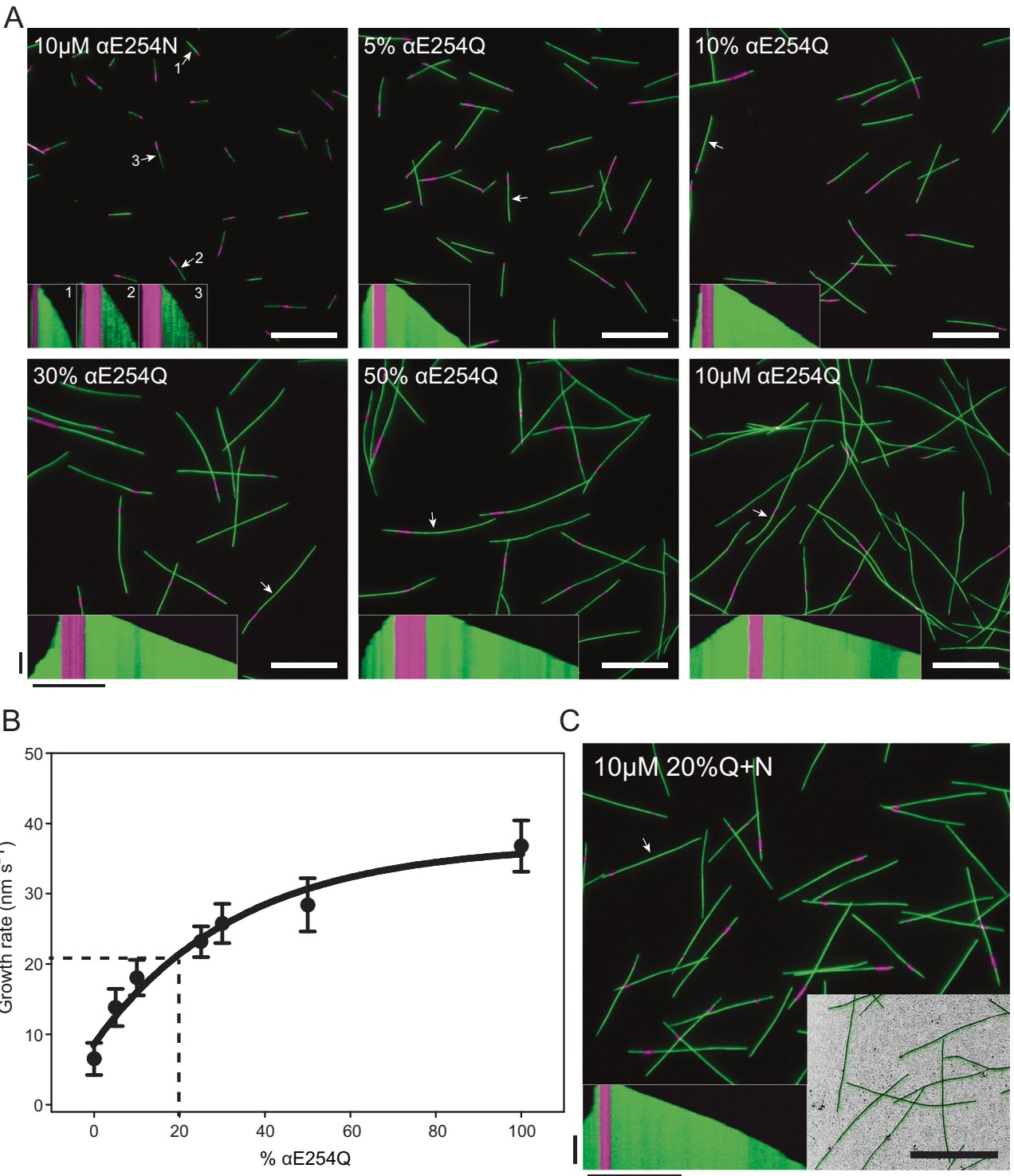

undecorated microtubules and microtubules decorated with an N-terminal fragment of EB3 consisting of its GTP-cap recognizing calponin homology (CH) domain[14,30] (Supplementary Fig. 6A–D).

After classifying microtubules into 3D classes (Supplementary Fig. 1E), we first analyzed the protofilament numbers of the different GTP-microtubules. Mammalian GDP microtubules polymerized in vitro from WT-tubulin in the same buffer used in our experiments and in the presence of GTP have typically mostly 13 protofilaments[42,43]. However, nucleotide analogs are known to alter the protofilament distribution. For example, GMPCPP-microtubules have mostly 14 protofilaments, whereas BeF$_3^-$GDP microtubules have

preferentially 12 protofilaments[27,29]. Similarly, buffer components can change protofilament number, for example glycerol induces 14-protofilament-microtubules[20]. We found that GTP-containing αE254Q-microtubules in a buffer without glycerol displayed mostly 12 and 13 protofilaments, whereas 20%Q + N-microtubules were predominantly composed of 13 protofilaments (Fig. 5A and Supplementary Table 2). This adds yet another piece of evidence that 20% Q + N-microtubules eliminate mutation-specific effects and faithfully resemble WT-GTP-microtubules. The addition of the EB3 calponin homology domain (EB3-CH) to the pre-polymerized microtubules had only a small effect on the protofilament numbers of the

**Fig. 3 | Mosaic microtubules consisting of mixtures of αE254Q and αE254N-tubulin. A** TIRF microscopy images of microtubules polymerized in the presence of αE254Q-tubulin, αE254N-tubulin or mixtures of both tubulins from immobilized CF640R-labeled GMPCPP-stabilized microtubule seeds (purple) in the presence of mGFP-EB3 (green). Percentage of αE254Q-tubulin in mosaic microtubules as indicated. Insets show example kymographs, with the corresponding particles indicated by white arrows. Images were obtained from different experiments with slightly different imaging conditions. To display them with similar contrast, the mean background (per pixel) in five separate regions and the mean EB3-GFP signal for five different microtubules were calculated. The contrast of each image was adjusted such that the minimum value equaled the mean background, and the maximum value equaled the mean EB3-GFP signal (see "Methods"). αE254N-microtubules display segments with pronounced differences in EB3 binding affinity, in contrast to microtubules containing αE254Q-tubulin. **B** Average growth rates of mosaic microtubules (as in (**A**)) as a function of the percentage of αE254Q-tubulin in the αE254Q/αE254N-tubulin mixture. $N = 2$ independent experiments; The number of growth events analyzed per condition (n) is: αE254E (WT)−54; 0% αE254Q−65; 5% αE254Q−74; 10% αE254Q−41; 25% αE254Q−51; 30% αE254Q−34; 50% αE254Q−42; 100% αE254Q−29. Error bars represent SEM. An empirical fit to the data (see "Methods") allows estimating the microtubule growth rate for a 20% Q + N-tubulin mixture (-21 nm/s, dashed lines). **C** TIRF microscopy image and an example kymograph of 20%Q + N-microtubules elongating from CF640R-labeled GMPCPP microtubule seeds (purple) in the presence of mGFP-EB3 (green). Bottom-right inset shows a combined IRM/TIRF microscopy image of 20%Q + N-microtubules (black), depicting homogeneous mGFP-EB3 binding. Total tubulin concentration in all experiments is 10 μM, mGFP-EB3 concentration is always 40 nM. White and black horizontal scale bars represent 10 μm, vertical scale bars represent 10 min. Source data are provided as a Source Data file.

decorated microtubules, confirming that EBs promote the polymerization of 13-protofilament microtubules only when present during polymerization[44].

Next, we generated cryo-EM 3D reconstructions of the four types of microtubule preparations, yielding unsymmetrized C1 and symmetrized electron density maps with a resolution of around 4.1 Å and 3.6 Å, respectively at the 0.143 FSC cutoff (Supplementary Fig. 6F, G and Supplementary Table 3)[45], which allowed the building of microtubule atomic models from the symmetrized electron density maps (Fig. 5B, C, Supplementary Figs. 7 and 8 and Supplementary Table 3). We determined the protofilament twist and dimer rise of the microtubule lattices from their unsymmetrized C1 microtubule maps (Table 3). We found that αE254Q-microtubules exhibit a dimer rise of 82.34 ± 0.00 Å, which indicates a slight microtubule lattice expansion when compared to compact, wild-type GDP microtubules (81.5 Å) (Supplementary Fig. 8A). They are less expanded than wild-type GMPCPP-microtubules (84.0 Å), or αE254A- (83.2 Å) or αE254N-microtubules (83.5 Å)[20,32]. Interestingly, αE254Q-microtubules show a mild positive protofilament twist of +0.11 ± 0.00°, similar to αE254N-microtubules ( + 0.13°), which is close to that of straight, wild-type GDP microtubules ( + 0.08°) but different from the pronounced negative twist of αE254A (−0.22°) and the pronounced right-handed twist of GMPCPP-microtubules ( + 0.23°)[30,32]. This analysis suggests that even a mild expansion of the microtubule lattice is enough for microtubule stabilization by αE254Q-tubulin and that a supertwist is not essential for conferring microtubule stability.

20% Q + N-microtubules show a dimer rise of 83.24 ± 0.01 Å which is more similar to that of previously reported expanded microtubule lattices while the protofilament twist of +0.10 ± 0.01° is still similar to that of wild-type GDP microtubules[32] (Fig. 5D). Hence, considering that 20%Q + N-microtubules are faithful mimics of the GTP-cap, our results indicate that the microtubule GTP cap exhibits an expanded lattice, which provides structural stability to the lattice while the protofilament twist does not change much after GTP hydrolysis.

Remarkably, the EB3-CH-decorated αE254Q and 20%Q + N-microtubule reconstructions preserve an expanded lattice (82.87 ± 0.00 and 82.89 ± 0.00, respectively) while displaying an induced negative twist (−0.23 ± 0.00° and −0.28 ± 0.03°, respectively) (Table 3, Fig. 5C, E, and Supplementary Fig. 8B). Considering that any previous structure of a microtubule decorated with EB3-CH showed both a compact and left-handed twisted lattice[20,30,32], we can conclude that the presence of αE254Q can block the lattice compaction of a GTP-microtubule lattice, even when only a fraction of tubulins carry this mutation as in mosaic 20%Q + N-microtubules.

## Discussion

In this study, we aimed to gain a deeper understanding of the biochemical and structural properties of the GTP cap using GTP hydrolysis-deficient, human tubulin mutants. All such mutants generated to date[31,32], including the αE254Q-tubulin reported in this study, which

features the smallest change possible, have their own characteristic, mutation-specific effects on the microtubule structure and biochemistry. For this reason, we developed a mosaic microtubule consisting of a mixture of mutant tubulins as an improved GTP-cap mimic. At an optimized 1:4 ratio of αE254Q and αE254N-tubulin, mosaic microtubules nucleated and grew as straight, 13-protofilament microtubules like WT microtubules, but were stable and bound EBs with high affinity due to the presence of GTP-tubulin throughout the microtubule lattice. Thus, we conclude that the 20%Q + N mosaic microtubule is a faithful GTP cap mimic, isolating the structural and biochemical consequences of the homogeneous presence of GTP in the E-site of tubulin dimers, eliminating any detectable other mutation-specific effects.

Different non-hydrolyzable GTP analogs as well as different mutations of the catalytic E254 residue in α-tubulin all cause microtubules to be stable. But they show distinct structural and biochemical properties, emphasizing how sensitive the microtubule lattice is at the site of GTP hydrolysis and how adaptable the GTP(-like) lattice conformation is (Fig. 6). It appears that very small changes at this site can be amplified by the cooperative nature of the microtubule lattice. Microtubule lattice cooperativity is especially evident in αE254N-microtubules, which have extended segments of differing lattice structures as indicated by different EB-binding affinities[32]. Notably, cooperative microtubule lattice changes have also been reported to be induced by small microtubule-binding molecules such as taxanes[46].

The success of the mosaic microtubule approach in eliminating mutation-specific effects by mixing different mutants indicates that breaking the repetition of always the same local structural change can "normalize" the microtubule structure, as indicated, for example, by the microtubule growth rate that becomes normal for mosaic microtubules at an optimized tubulin mutant mixing ratio. Interestingly, the growth rates of GTP hydrolysis-deficient mosaic microtubules did not increase linearly with the proportion of αE254Q-tubulin in the mixture. This finding indicates that not only the properties of the soluble tubulin dimers determine microtubule growth speed but also properties of the growing microtubule end itself that may affect the kinetics of tubulin addition or conformational changes of tubulins as they incorporate into the lattice. The positive deviation from a linear relationship that is maximal around the optimal tubulin mutant mixing ratio (Fig. 3B) could indicate that the growing end structure of straight, 13-protofilament 20%Q + N mosaic microtubules is in fact optimal for tubulin incorporation, suggesting that the end structure of the natural GTP cap is optimal for growth.

The lattice parameters of 20%Q + N mosaic microtubules predict that the GTP cap displays an expanded lattice with hardly any supertwist, very similar to compact wild-type GDP microtubules[19,20,32]. Hence, this finding suggests that the net structural change induced by GTP hydrolysis in mammalian microtubules is mainly an axial compaction of tubulin. However, the relatively slow GTP hydrolysis rate in the second range, as estimated from the lifetime of the high-affinity EB-binding sites at growing WT-microtubule ends and from inorganic

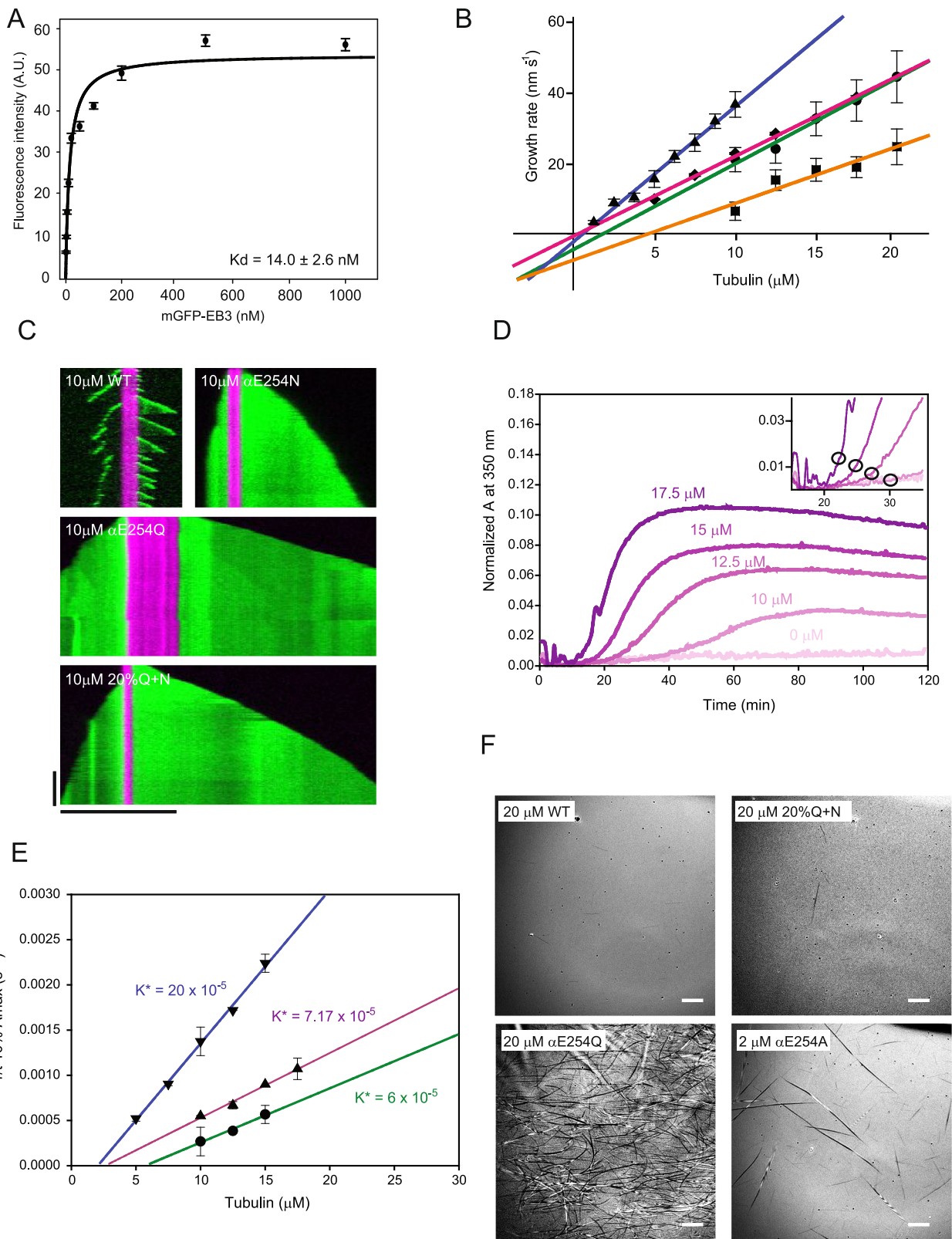

phosphate release kinetics[11,31,47], suggests that there is a considerable energy barrier for compaction and GTP hydrolysis. EB proteins, which are known to accelerate this transition[9,24], therefore help to circumvent this energy barrier separating the GTP- and GDP-microtubule lattice states.

EB proteins were observed to induce compaction and a negative supertwist in all previous EB-decorated microtubule structures solved

to date (Fig. 6)[20,30,32]. We report here two EB3-CH-decorated structures with a negative twist, but an expanded lattice conformation. This expansion means that the presence of a glutamine at residue position α254 at the catalytic site can inhibit compaction, possibly arresting the microtubule lattice on its pathway toward the GDP-microtubule lattice state. In WT microtubules, a negative protofilament twist may be transiently required to facilitate compaction of the GTP lattice to allow

**Fig. 4 | Biochemical characterization of mosaic 20%Q + N-microtubules. A** Plot of average maximum mGFP-EB3 fluorescence intensities measured along 20% Q + N-microtubules at different mGFP-EB3 concentrations. $N = 2$ independent experiments, $n = 100$ microtubules per mGFP-EB3 concentration. Error bars represent SEM. A hyperbolic fit yields the dissociation constant, $K_d$, as indicated. Fluorescence intensities are represented in arbitrary units (A.U.). **B** Plot of average microtubule growth rates at different total tubulin concentrations in the presence of 40 nM mGFP-EB3 and the respective linear regressions for αE254E (WT, circles, green line), αE254Q (triangles, blue line), αE254N (squares, orange line), and 20% Q + N-microtubules (diamonds, purple line). $N = 3$ independent experiments; a total of 29–362 growth events were analyzed per condition. Error bars represent SEM. From the slope and the intercept, we determined the apparent on-rate constant $k_{on,app}$ and off-rate constant $k_{off,app}$, respectively, as summarized in Supplementary Table 1. **C** TIRF microscopy kymographs of αE254E (WT), αE254N, αE254Q, and mosaic 20%Q + N-microtubules for the 10 μM tubulin condition in (**B**), polymerized from stabilized seeds (purple) in the presence of 40 nM mGFP-EB3 (green). Horizontal and vertical scale bars represent 10 μm and 5 min, respectively. **D** Time course of the turbidity after start of polymerization of 20%Q + N-microtubules at different total tubulin concentrations as indicated. The inset shows a magnification of the beginning of the polymerization reaction. Black circles indicate where the absorbance reaches 10% of its maximum value (10% Amax) which is used to calculate the apparent self-nucleation rate constant. **E** Plots to determine the apparent self-nucleation rate constant $K^*$ of WT (circles, green linear regression), αE254Q (triangles-down, blue linear regression), and 20%Q + N-tubulin (triangles-up, purple linear regression). $N = 3$ independent experiments; $n = 2$ replicas per condition. Error bars represent SEM. **F** IRM images of microtubules spontaneously nucleated in solution from different tubulins, as indicated, at the concentrations, as indicated. Scale bars represent 10 μm. Source data are provided as a Source Data file.

for efficient GTP hydrolysis[48]. This view is also in agreement with the structure of BeF$_3^-$-GDP microtubules that was recently described as a compact, left-handed supertwisted GTP state[29], which could be thus interpreted as a pre-hydrolysis GTP state. In conclusion, axial lattice expansion appears to be a characteristic feature of the stable GTP cap, while supertwist variations may rather be needed transiently on the pathway to lattice compaction of mammalian microtubules and GTP hydrolysis.

In conclusion, our results provide insights into the structure of the GTP cap and the structural transitions during GTP hydrolysis. The GTP hydrolysis-deficient mosaic microtubules as the currently most faithful GTP cap mimic promise to be a useful tool for future studies into the remaining puzzles of the structure-stability relationship of the GTP cap. Moreover, the ability to generate mosaic microtubules from recombinant tubulins can be seen as a more broadly useful research tool, for example, for the investigation of the compositional complexity of microtubules with respect to different tubulin isotypes[49–52] or different post-translational modifications[53,54].

## Methods

### Bacteria and insect cells

Bacteria (*E. coli* DH5α, BL-21 (DE3), pRiL, DH10Bac) were cultured in Luria Bertani (LB) broth supplemented with required antibiotics. Insect cells were grown in suspension shaking at 120 rpm at 27 °C, either in Sf-900 III SFM media (12659017, Gibco) for *S. frugiperda* Sf21 cells (12682019, Gibco), or in ExpressFive SFM media (10486025, Gibco) supplemented with 16 mM L-glutamine (25030081, Gibco) for *Trichoplusia ni* High Five cells (B85502, Gibco).

### Molecular cloning

The construct TUBA1B-intHis6 TUBB3-Gly2-Ser-Gly2-TEVsite-StrepTagII (pJR374) was modified by Quickchange mutagenesis with the following primers: Fw 5′-CAACGTGGACCTCACTCAGTTCCAAAC-CAACCTGG and Rv: 5′-CCAGGTTGGTTTGGAACTGAGTGAGGTC-CACGTTG, to generate the expression construct TUBA1B_E254Q-intHis6 TUBB3-Gly2-Ser-Gly2-TEVsite-StrepTagII (pRG01, αE254Q) for hydrolysis-deficient human tubulin dimers, called here αE254Q-tubulin. Baculovirus preparation for αE254Q expression was carried out according to the manufacturer's protocol (Bac-to-Bac system, Life Technologies). Briefly, the expression construct was transformed into DH10Bac *E. coli*, and successful transposition was confirmed by PCR screening. The recombinant bacmid DNA was then purified and transfected into Sf21 cells, where the virus was produced and harvested from the supernatant. This virus was used to infect fresh insect cells at a low multiplicity of infection (MOI) for amplification. Through three rounds of infection, high-titer baculovirus stocks were obtained. Baculovirus-infected insect cells (BIICs) were prepared and frozen for storage at −80 °C as previously described[55]. Briefly, High Five cells were infected with baculovirus at a low multiplicity of infection (MOI) and harvested 48 h post-infection. The harvested BIICs were resuspended in a freezing medium containing 10% DMSO, frozen, and stored at −80 °C for long-term use.

### Protein expression and purification

αE254Q-tubulin expression was induced by adding 3 ml of frozen BIICs per liter to a High Five insect cell culture grown to densities of ~1.2 × 10⁶ cells/ml. Cells were harvested 48 h post-infection by centrifugation (15 min, 900× *g*, 4 °C). Cell pellets were then washed with cold PBS, centrifuged again (15 min, 1000× *g*, 4 °C) and stored at −80 °C.

Recombinant human tubulin was purified following a previously described protocol[31,56] using slightly modified buffers. Cell pellets obtained from 2 L of High Five insect cells expressing human TUBB3-TEVsite-StrepTagII, with the C-terminal StrepTagII cleavable by tobacco etch virus (TEV) protease, and TUBA1B-His, with internal 6x His-Tag (WT, αE254A, αE254N, and αE254Q) were resuspended 1:1 (vol/vol) in cold lysis buffer (80 mM PIPES, 1 mM EGTA, 1 mM MgCl₂, 50 mM imidazole, 100 mM KCl, 0.2 mM GTP, 1 mM β-ME, pH 7.2) and lysed using an Emulsiflex-C5 homogenizer (Avestin). The resulting lysate was clarified by ultracentrifugation (160,000× *g*, 45 min, 4 °C), filtered using 0.22 μm PES syringe filters (4602, Merck) and flowed through a 5-mL HisTrap HP column (GE17-5255-01, Cytiva). The column was then first washed with 5 column volumes (CVs) of Ni-dilution buffer (80 mM PIPES, 1 mM EGTA, 6 mM MgCl₂, 50 mM imidazole, 0.2 mM GTP, 1 mM β-ME, pH 7.2), then with 5 CVs of Ni wash buffer 1 (80 mM PIPES, 1 mM EGTA, 11 mM MgCl₂, 0.2 mM GTP, 5 mM ATP, 1 mM β-ME, pH 7.2), then with 5 CVs of Ni wash buffer 2 (80 mM PIPES, 1 mM EGTA, 5 mM MgCl₂, 0.1% (vol/vol) Tween-20, 10% (w/vol) glycerol), 0.2 mM GTP, 1 mM β-ME, pH 7.2) and finally with 5 CVs of Ni-dilution buffer. The protein was eluted with Ni-elution buffer (80 mM PIPES, 1 mM EGTA, 5 mM MgCl₂, 300 mM imidazole 0.2 mM GTP, 1 mM β-ME, pH 7.2). The eluted protein was collected in a 96-well plate containing 1 volume of Strep binding buffer (80 mM PIPES, 1 mM EGTA, 5 mM MgCl₂, 0.2 mM GTP, 1 mM β-ME, pH 7.2) in order to immediately dilute 4 times the imidazole present in the Ni-elution buffer. The eluted fractions corresponding to the main peak were joined, diluted 2 times with Strep binding buffer, and passed through two HiTrap SP FF columns of 1 mL (17505401, Cytiva) and one StrepTrap HP of 5 mL (28-9075-48, Cytiva) serially connected and equilibrated in Strep binding buffer. The columns were then washed with 2 CVs of Strep binding buffer, and the protein was eluted in Strep elution buffer (80 mM PIPES, 1 mM EGTA, 5 mM MgCl₂, 2.5 mM D-desthiobiotin, 0.2 mM GTP, 1 mM β-ME, pH 7.2). The eluted protein was supplemented with TEV protease at a ratio of 1:10 (1 mg of TEV protease per 10 mg of tubulin) and incubated for 2 h at 4 °C in order to remove the C-terminal Strep-Tag II from TUBB3. After the tag cleavage, the protein was centrifuged at 204,428× *g*, 10 min, 4 °C to remove possible aggregates and insoluble fractions. The supernatant was then passed through two 1 ml HiTrap SP FF columns of 1 mL (17505401,Cytiva) serially connected to the top of two HiPrep Desalting 26/10 desalting columns (17-5087-01,

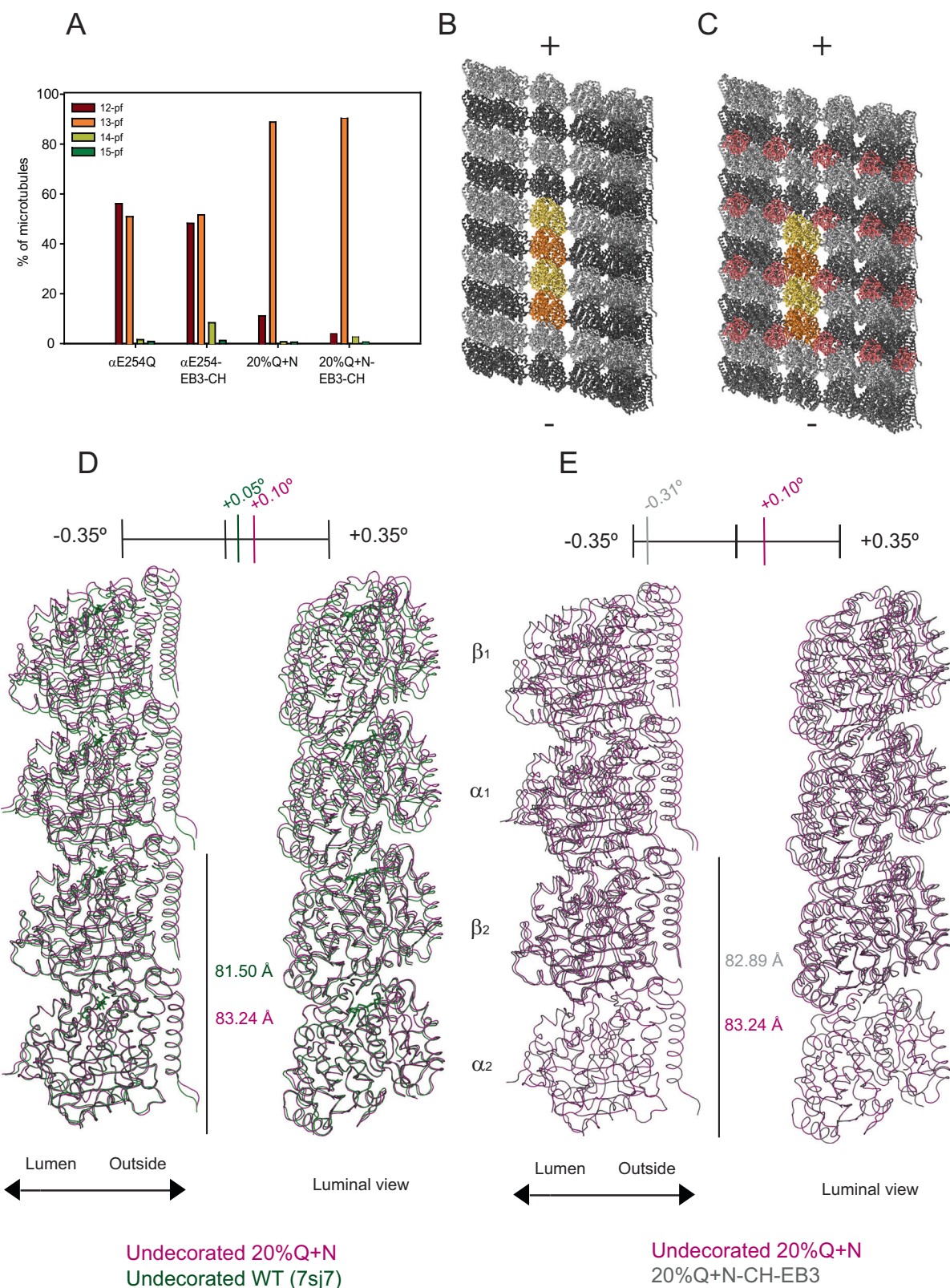

**Fig. 5 | Cryo-EM analysis of 20%Q + N-microtubules. A** Histogram of the relative abundance of 12- to 15-protofilament (pf) microtubule specimens observed in cryo-EM datasets (see also Supplementary Table 2). Plus (+) and minus (−) ends are indicated. **B**, **C** Cryo-EM structures of undecorated (**B**) and EB3-CH-decorated (**C**) 20%Q + N-microtubules. **D**, **E** Atomic model building of mosaic microtubules from their corresponding symmetrized maps. Comparison of the Cα traces of two consecutive dimers after model superimposition on the intermediate domain of the α2-tubulin chain. **D** Comparison between undecorated 20%Q + N-microtubules (magenta) and undecorated WT microtubules (green) (PDB: 7SJ7)[32]. **E** Comparison between undecorated 20%Q + N-microtubules (magenta) and CH-EB3-decorated 20%Q + N-microtubules (gray). The dimer rise (next to bottom tubulin dimers) and protofilament twist (at the top) as determined from the C1 maps using RELION are indicated.

**Table 3 | Lattice parameters of 13-protofilament microtubules**

|  | αE254Q | αE254Q-EB3 | 20%Q + N | 20%Q + N-EB3 |
|---|---|---|---|---|
| Protofilament twist (°) | 0.11 ± 0.00 | −0.23 ± 0.00 | 0.10 ± 0.01 | −0.28 ± 0.03 |
| Dimer rise (Å) | 82.34 ± 0.00 | 82.87 ± 0.00 | 83.24 ± 0.01 | 82.89 ± 0.00 |

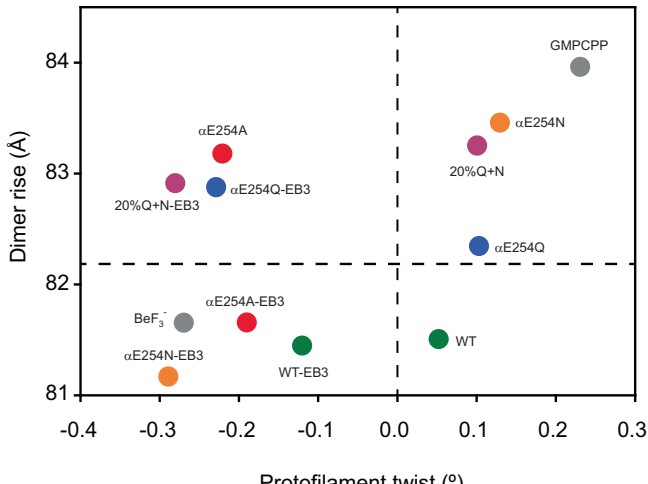

**Fig. 6 | Lattice parameters of WT and microtubule GTP cap mimics.** Protofilament twist versus dimer rise plot of αE254Q, αE254Q-EB3-CH, 20% + N, and 20% Q + N-EB3-CH-microtubules compared with previous undecorated and EB3-CH-decorated GTP-cap mimics and GDP microtubules[34]. Data for the structures characterized in this work are shown in Table 3 and Supplementary Tables 2 and 3. Source data are provided as a Source Data file.

Cytiva) to remove the TEV protease and to equilibrate the protein in tubulin storage buffer (80 mM PIPES, 1 mM EGTA 1 mM MgCl₂, 0.2 mM GTP, pH 6.8). The tubulin-containing fractions were pooled, concentrated using Amicon Ultra-4, 50 kDa MWCO (Millipore) to a concentration higher than 5 mg/mL and ultracentrifuged (280,000× $g$, 10 min, 4 °C). The resulting clarified pure tubulin solution was aliquoted, snap-frozen, and stored in liquid nitrogen for next use.

Porcine brain tubulin was purified as previously described[57]. Briefly, tubulin was purified from brain extract by sequential cycles of polymerization of tubulin at 37 °C in high-concentration PIPES buffer supplemented with ATP, GTP, and glycerol, followed by depolymerization on ice. For long-term storage 5-ml-tubulin aliquots were frozen and stored in liquid nitrogen. Tubulin used in experiments was passed through another polymerization/depolymerization cycle and was snap-frozen and stored in 10 μl aliquots in liquid nitrogen. Tubulin was labeled with CF640R-NHS (SCJ4600044, Merck) or biotin-NHS (203112, Merck), as previously described[58]. Briefly, NHS-fluorophores dissolved in anhydrous DMSO were added drop by drop to microtubules obtained from cycled tubulin while gently mixing. After 40 min, the labeled microtubules were depolymerized at 4 °C. The samples were then ultracentrifuged (280,000× $g$, 10 min, 4 °C), and the pellet was discarded. The supernatant containing labeled tubulin was aliquoted, snap-frozen, and stored in liquid nitrogen in 5 μl aliquots.

mGFP-EB3 used for TIRF microscopy experiments was purified as previously described[31]. Briefly, mGFP-EB3 was expressed in *E. coli* BL-21 cells and purified by nickel-affinity chromatography, followed by size exclusion chromatography. Monomeric human EB3 calponin homology (CH) domain (residues 1-200) used for cryo-EM experiments was purified as previously described[30,32]. Briefly, EB3-CH was expressed in *E. coli* BL-21 cells and was purified by nickel-affinity chromatography, followed by size exclusion chromatography.

## GMPCPP microtubule seed preparation
GMPCPP seeds were prepared as described[58]. Briefly, 6.67 μM of non-labeled porcine brain tubulin, 5 μM biotinylated porcine brain tubulin and 5 μM CF640R-labeled porcine brain tubulin, were mixed with 0.5 mM GMPCPP (NU-405L, Jena Bioscience), diluted up to a final volume of 60 μL with BRB80 (80 mM PIPES, 1 mM EGTA, 1 mM MgCl₂, pH 6.8) and incubated 5 min on ice. This tubulin mix was then incubated 45 min at 37 °C to allow for polymerization. The seeds were diluted to a final volume of 500 μL and centrifuged at 17,000× $g$ at room temperature. The pellet was resuspended in 500 μL of BRB80, 10% glycerol. The resulting seed suspension was aliquoted, snap-frozen, and stored in liquid nitrogen until use. The final concentration of seeds, assuming a 50% polymerization efficiency would be 1 μM (polymerized tubulin dimers) which was used to calculate further dilutions.

## Determination of nucleotide content in microtubules
To determine the microtubule nucleotide content, microtubules were polymerized from 0.15 μM GMPCPP microtubule seeds in the presence of 3.4 M glycerol, 20 μM recombinant tubulin, and 1 mM GTP. Nucleotide was extracted as previously described[31]. Briefly, microtubule suspensions were treated with HClO₄ to a final concentration of 0.7% (w/vol) followed by immediate neutralization of the reaction by addition of sodium acetate to a final concentration of 200 mM. Then, the final sample was prepared by centrifugation at 15,000× $g$ for 5 min at 4 °C. Microtubule nucleotide content was determined by ion-pair HPLC chromatography, following a separation protocol based on a previously described procedure[59,60]. Briefly, the separation was performed using a C18 ZORBAX Eclipse Plus C18 (4.6 × 250 mm, 5 μm particle size, 95 A pore size) (AG959941-902, Agilent) previously equilibrated overnight at a 0.2 mL/min flow with ion-pair buffer (0.2 M K₂HPO₄, 0.1 M acetic acid, 4 mM tetrabutylammonium bromide, pH 6.65). Before injecting, samples were diluted 1:1 with ion-pair buffer and filtered using 0.22 μM centrifuge filters TUBE SPIN-X (CLS8161, Merck). The separation was performed with ion-pair buffer in 20 min isocratic runs at a flow of 1 mL/min. Nucleotide chromatograms were recorded by measuring the absorbance at 254 nm.

## TIRF microscopy and IRM assays
**Flow chamber preparation.** For TIRF microscopy experiments, flow chambers were assembled using one biotin-polyethylene glycol (PEG)-functionalized coverslip (Menzel 1.5H; Marienfeld High Precision 1.5H) and one poly-(L-lysine)-PEG-passivated glass slide separated by a double sticky tape, as described[61]. For combined IRM/TIRF microscopy experiments, flow chambers were built from two passivated biotin-PEG-functionalized coverslips separated by Parafilm to form channels as described[62,63], and the chamber was mounted on a custom-made sample stand for imaging.

Both types of flow chambers were further passivated for 10 min at room temperature with 5% Pluronic F-127 (P2443, Merck) and then washed twice with assay buffer (80 mM PIPES, 1 mM EGTA, 1 mM MgCl₂, 30 mM KCl, 1 mM GTP, 5 mM β-ME, 0.15% (w/vol) methylcellulose (4000 cP, Merck), and 1% (w/vol) glucose, pH 6.8) supplemented with 50 μg/mL κ-casein (C0406, Merck). Then, 50 μg/mL Neutravidin (A2666, Invitrogen) in assay buffer supplemented with 50 μg/mL κ-casein was flowed into the chamber and incubated for 3 min at room temperature and washed out twice with assay buffer. Next, biotinylated and CF640R-labeled GMPCPP microtubule seeds diluted to

3–5 nM (polymerized tubulin) in BRB80 were flowed in and incubated for 3 min at room temperature. After two final washes with assay buffer, the final samples were flowed into the chamber.

**Final sample preparation.** For assays with dynamic microtubules, samples were prepared as previously described[31,32]. Briefly, recombinant human tubulins and 40 nM mGFP-EB3 were diluted in assay buffer, supplemented with oxygen scavengers (180 mg/mL catalase (C40, Merck) and 752 mg/mL glucose oxidase (22778.01, Serva)). Samples flowed into the chamber, which was then sealed with silicone grease and placed on the microscope equipped with an incubator kept at 30 °C. Imaging started after 1 min.

To measure the mGFP-EB3 binding affinity to microtubules, microtubules were grown in a test tube from 0.5 μM biotinylated and CF640R-labeled GMPCPP microtubule seeds in the presence of 20 μM αE254Q or 20%Q + N-tubulin and 1 mM GTP for 1 h at 37 °C in BRB80. This microtubule suspension was diluted 1:100 with warm BRB80 buffer containing different concentrations of mGFP-EB3 and oxygen scavengers (180 mg/mL catalase and 752 mg/mL glucose oxidase) and then flowed into a TIRF microscopy flow chamber prepared as above described, directly after washing out the Neutravidin solution twice with assay buffer. The flow chamber was then sealed with silicone grease and placed on the microscope equipped with an incubator kept at 30 °C. Imaging started after 1 min.

To visualize spontaneous tubulin nucleation in solution, microtubules were grown in a test tube from tubulin at the specific concentration for each condition in the presence of 1 mM GTP for 1 h at 37 °C in BRB80. This microtubule suspension directly flowed into a non-passivated flow chamber, which was only washed twice with BRB80. The flow chamber was then sealed with silicone grease and placed on the microscope equipped with an incubator kept at 30 °C. Imaging started after 1 min.

**TIRF microscopy imaging.** TIRF microscopy assays with dynamic microtubules and mGFP-EB3 were performed using a Nikon-TIRF microscope (Ti-E, Cairn Research, United Kingdom). A 100× oil-immersion TIRF objective (Nikon CFI SR Apo, NA = 1.49) was utilized to focus collimated beams from a 488 nm and 640 nm diode laser onto the backfocal plane. The beams were azimuthally scanned using a Galvo scanning system (iLas2, Gataca Systems, France). A quadband dichroic mirror (Chroma: ZT405/488/561/640), 488 nm band-pass filter (Chroma: ET525/50) and 655 nm long-pass filter (Chroma: ET655LP) were used to separate incident and emitted light. Both channels were simultaneously recorded using two Andor iXon 888 Ultra EMCCD cameras (Andor Technology, United Kingdom) controlled by Meta-Morph software (Molecular Devices, USA). Time-lapse movies were recorded with a frame rate of 10 Hz and an exposure time of 50 ms for both channels. The duration of the movies ranged from 20 to 60 min. For persistence length calculations, snapshots of different fields of view were acquired after 20 min. For mGFP-EB3 binding curve measurements, 10 snapshots were acquired for each condition.

**Combined IRM/TIRF microscopy imaging.** Interference reflection microscopy (IRM) was implemented on the Nikon-TIRF microscope based on a published design (Supplementary Fig. 9A)[64]. We replaced the collimation lenses and aperture iris (L1, L2, L3, AI in Simmert et al., Fig. 2) by a fiber-coupled LED at 780 nm (M780F2; Thorlabs, USA) with a multi-mode fiber (M28L01; Thorlabs) and large-beam fiber collimator (C40SMA-B; Thorlabs). The collimator, a field iris (SM1D12; Thorlabs) and an XY translation mount (CXY1A; Thorlabs) were mounted in a Ø1" lens tube and placed at the EPI illumination port of the iLas2 adapter (Supplementary Fig. 9B; Cairn Research, UK). The beam was aligned via the XY translation mount onto a 700 nm short-pass dichroic (FF700-SDi01-25×36; Semrock, USA) in the iLas2 adapter to combine both

IRM imaging at 780 nm and TIRF microscopy imaging with the 488 nm laser. The IRM and TIRF beams were ultimately focused in the backfocal plane via a lens in the iLas2 adapter. IRM images were captured using a 50–50 dichroic, the 100× objective, the 655 nm long-pass filter and the Andor iXon 888 Ultra EMCCD (see TIRF microscopy imaging). This setup maximized the signal-to-noise ratio of IRM imaging, at the expense of a reduction of mGFP signal, compared to TIRF-only microscopy imaging. For combined IRM/TIRF time-lapse imaging (Supplementary Fig. 9C), 40 IRM images were recorded in fast-stream acquisition mode, 1 TIRF frame (488 nm excitation) was acquired, and then again 40 IRM images. All 80 IRM images were averaged to a single frame, resulting in one TIRF and one IRM image at approximately the same acquisition time. This process was repeated yielding a 20 min time-lapse movie for both IRM and TIRF microscopy with a 15 s time. For IRM imaging of microtubules nucleated in solution, a single image was obtaining from averaging 80 IRM images and no TIRF image was acquired. To perform background subtraction for the IRM time-lapse images and single images, an additional set of 101 IRM frames was acquired after each movie while moving the stage automatically in zig-zag by 1 μm in the $xy$-plane. The median of this set of images represented the background and was subtracted from all the averaged IRM time-lapse images to obtain the final IRM data.

## TIRF microscopy image data analysis

The first step was to align the two channels of each image in MATLAB (MathWorks, USA) using a calibration slide (Argo-HM, Argolight, France). All the subsequent processing and analysis steps were performed employing Image J 1.53t (NIH, USA) by employing custom-made macros. Following channel alignment, images were then drift-corrected, and the background was subtracted using a rolling-ball algorithm with a radius of 50 pixels.

To adjust the contrast of images obtained from different experiments with slightly different imaging conditions (Fig. 3), FIESTA[65] was used to measure the mean background (per pixel) in five separate regions and the mean EB3-GFP signal for 5 different microtubules. The mean background was determined by drawing five random rectangular regions and calculating the mean of each region. The mean EB3-GFP signal was determined by drawing linescans perpendicular to 5 randomly selected bright microtubules. The 19 pixels wide linescans were averaged and then fitted with a one-dimensional Gaussian function The contrast of each image was adjusted such that the minimum value equaled the mean background, and the maximum value equaled the mean EB3-GFP signal, which can be calculated by the sum of the mean background and mean amplitude of the Gaussian fits. For the kymographs in Fig. 3, the maximum contrast value was increased by 25% compared to the individual images (multiplied by 1.25) to allow for better visualization of slight intensity variations.

To quantify the EB3-GFP signal on the microtubules (Supplementary Fig. 3), FIESTA was used to create 19 pixels wide linescan along the microtubules. The background was subtracted from the integrated intensity along the linescan. The average EB3-GFP signal per microtubule was determined only from plus-segments of the microtubules excluding seeds and microtubule crossings, and for WT GDP microtubules also excluding the GTP cap region at microtubule plus ends.

To generate kymographs, segmented lines, 11 pixels wide, were manually drawn on a maximum-intensity projection image along the filaments, extending from the seeds. Kymographs were then generated from the movie stack along each line. The end positions of each filament were manually tracked to obtain growth rates. Statistics of growth speed were obtained using JASP 0.16.3 (University of Amsterdam).

To obtain an expression describing the dependence of the growth speeds of mosaic microtubules on tubulin mutant composition, we

fitted an empirically chosen exponential growth to maximum function to the measured growth speeds plotted as a function of αE254Q content, using SigmaPlot 11.0 software (RRID:SCR_003210). Using this fit, we determined the composition at which mosaic microtubules grow with WT growth speed.

The apparent association constant $k_{on}$ and dissociation constant $k_{off}$ (of tubulins at growing microtubule ends was obtained from a linear regression to measured growth rates plotted as function of total tubulin concentration[35,36]:

$$Vg = (8\,\text{nm}/13)(k_{on,app}C - k_{off,app}) \qquad (1)$$

Where $v_g$ is the growth rate of a 13-protofilament microtubule at a given tubulin concentration c, considering that a tubulin is 8 nm long.

To determine the affinity of mGFP-EB3 for αE254Q and 20%Q + N mosaic microtubules, the maximum mGFP-EB3 fluorescence intensities for each analyzed microtubule was determined using Image J 1.53t (NIH, USA). The mGFP-EB3 dissociation constant (Kd) was determined by fitting a one-site binding model to the averages of the maximum intensities plotted as a function of mGFP-EB3 concentration using SigmaPlot 11.0 (RRID:SCR_003210),

$$I = \frac{I_{max}c}{I_{max} + c} \qquad (2)$$

where $I$ is the measured average intensity for each mGFP-EB3 concentration $c$ and $I_{max}$ is the maximum intensity at saturation of binding.

The average persistence length of a population of microtubules was calculated from the microtubule curvature distribution, as described[40]. Briefly, for αE254Q and 20%Q + N-microtubules, snapshots of microtubules were acquired in the mGFP-EB3 channel in five different randomly selected fields of view. For WT microtubules, 3 independent experiments were performed and the microtubule contours in one field of view each were obtained from the Z-projection of the mGFP-EB3 comet fluorescence intensities. Thirty contours were traced for each condition using JFilament software, and the Persistence Length Analyzer software[40] was utilized to apply a curvature distribution analysis to each set of contours. Curvatures were measured between sample points in the contours in a recursive manner, subsampling 20 different combinations of distances between points (subsampling levels ranging from m = 1 to m = 20). Due to the relatively small size of our datasets, we applied bootstrapping methods with a total of 100 iterations to account for sampling variability.

## LC-MS/MS mass spectrometry

20%Q + N-microtubules were polymerized in BRB80 from a mixture of 4 μM αE254Q, 16 μM αE254N-tubulin (20 μM total tubulin) and 1 mM GTP at 37 °C. The concentration of the resulting microtubule suspension was measured using a NanoDrop™One/One (Thermo Fisher Scientific) spectrophotometer and the corresponding volume for 20 μg of microtubules was centrifuged at 17,000× g at room temperature. The supernatant was discarded, and the pellet was washed twice with warm BRB80 and stored at −20 °C until further analysis. In parallel, we prepared calibration samples containing 20 μg of unpolymerized tubulin at different αE254Q: αE254N ratios: 0:100, 5:95, 10:90, 20:80, 35:75, and 50:50. Two technical replicates for each microtubule sample and tubulin calibration solution were processed for each experiment. Experiments were performed in duplicate, with two independent biological replicates (n = 2). Microtubule pellet and tubulin solution samples were treated and trypsin-digested as described[66]. Briefly, samples were reduced with 30 nmol dithiothreitol at 37 °C for 60 min, followed by alkylation in the dark with 60 nmol iodoacetamide at 25 °C for 30 min. The protein extract was then diluted to 2 M urea using 200 mM ammonium bicarbonate and digested with endoproteinase

LysC (1:10 w) at 37 °C for 6 h (Wako, cat. no. 129−02541). Subsequently, the solution was further diluted twofold with 200 mM ammonium bicarbonate for overnight trypsin digestion (1:10 w) at 37 °C (Promega, cat. no. V5113). After digestion, the peptide mixture was acidified with formic acid and desalted using a MicroSpin C18 column (The Nest Group) before LC-MS/MS analysis. For LC-MS/MS analysis, a chromatographic separation was performed over a 90-min gradient from 5% to 40% solvent B, with solvent A being 0.1% formic acid in water and solvent B being 0.1% formic acid in 80% acetonitrile. MS data were acquired as described[67]. Briefly, positive ionization mode with data-dependent acquisition was performed using an Orbitrap Fusion Lumos mass spectrometer (Thermo Fisher Scientific) coupled to an EASY-nLC 1200 (Thermo Fisher Scientific) for full MS scans (m/z 350–1400, resolution 120,000) and an ion trap for fragment spectra via HCD (28% collision energy). Dynamic exclusion was set to 60 s, with a Top Speed algorithm for precursor selection. Instrument performance was monitored with digested BSA and QCloud[68].

The acquired mass spectra were analyzed using the Proteome Discoverer software suite (v2.5, Thermo Fisher Scientific) and the Mascot search engine (v2.6, Matrix Science)[67]. The data were searched against a Swiss-Prot human database (as in April 2022, 20401 entries), α-tubulin E254Q and E254N, and a list of common contaminants and all the corresponding decoy entries[69]. For peptide identification a precursor ion mass tolerance of 7 ppm was used for MS1 level, trypsin was chosen as enzyme, and up to three missed cleavages were allowed. The fragment ion mass tolerance was set to 0.5 Da for MS2 spectra. Oxidation of methionine and N-terminal protein acetylation were used as variable modifications whereas carbamidomethylation on cysteins was set as a fixed modification. False discovery rate (FDR) in peptide identification was set to a maximum of 1%.

Skyline-daily software (22.2.1.352)[70] was used to create the library (output of Proteome Discoverer) and to extract the precursor area of unique peptides for α-tubulin αE254Q and αE254N, FDGALNVDLTQFQTNLVPYPR and FDGALNVDLTNFQTNLVPYPR, respectively, used for the determination of the αE254Q: αE254N ratios in the sample. The raw proteomics data have been deposited to the PRIDE[71] repository with the dataset identifier PXD058083.

## Turbidity assay to measure bulk tubulin polymerization kinetics

The time course of bulk tubulin polymerization was followed by measuring the turbidity of the suspension at 350 nm in Falcon 96-well polystyrene plates (10334791, Fischer Scientific) that were kept at 37 °C throughout the experiment and read in an Infinite M200 plate reader (TECAN). Samples with different concentrations of tubulin were prepared in BRB80 supplemented with 1 mM GTP and 3.4 M glycerol. A total of 120 time points at 30 s intervals were measured per sample. For controls, samples were recorded in the absence of tubulin.

Absorbance-over-time data recorded in turbidity assays were background corrected by subtracting the minimum absorbance value. Curve plotting and data analysis were performed using SigmaPlot 11.0 software (RRID: SCR_003210). We assessed the nucleation ability of each tubulin solution as described[39]. Briefly, the inverse of the time required to reach 10% of the maximum absorbance (1/t (10% Amax)) was plotted as a function of tubulin concentration and the apparent nucleation rate was obtained from the slope of a linear regression to these data.

## Cryo-EM of microtubules

**Microtubule sample and grid preparation.** For cryo-EM grid preparation, microtubules were polymerized in solution prior to their adsorption to the grid. To prepare αE254Q-microtubules, a 10 μL aliquot of αE254Q-tubulin was diluted with BRB80 buffer to a final concentration of 2 mg/mL. This protein solution was then centrifuged at 16,000×g at 4 °C for 10 min to pellet down any aggregates. The supernatant was then supplemented with 1 mM GTP and incubated at

37 °C for 45 min to induce tubulin polymerization. The subsequent microtubule suspension was centrifuged at 16,000×g and at 37 °C for 20 min to sediment microtubules. The pellet was then gently resuspended in the required volume of warm BRB80 buffer to prepare a final suspension at 2 mg/mL, which was finally incubated for 2 h at room temperature before preparing cryo-EM grids.

To prepare undecorated 20%Q + N-microtubules, a self-seeded assembly protocol was followed as previously described[72]. Briefly, we prepared microtubules as described in the previous paragraph using 10 μL of a mixture of 8 μM αE254Q and 32 μM αE254N-tubulin. After microtubule polymerization, the resulting suspension was sonicated for 2 min and pipetted up and down vigorously to generate short microtubule seeds. 30 μL of a mixture of 4 μM αE254Q and 16 μM αE254N-tubulin was supplemented at room temperature with 3 μL of seed suspension and polymerized at 37 °C as described for αE254Q-microtubules.

To obtain cryo-EM datasets, we employed Cflat R 2–2 grids. Prior to sample adsorption, grids were plasma cleaned for 1 min using a GloQube® glow discharge system (Quorum Technologies). Sample adsorption and vitrification were performed using a Vitrobot Mark IV vitrification system (Thermo Fischer) equilibrated at 37 °C and 100% relative humidity. For undecorated microtubules, 3 μL of αE254Q- or 20%Q + N-microtubule samples were directly added to the grid and immediately blotted for 3 s, followed by plunge freezing in liquid ethane. For EB3-CH-decorated microtubules, 3 μL of αE254Q- or 20% Q + N-microtubule samples were deposited on the grid and before blotting and freezing as described above, two additions of 3 μL of a 50 μM EB3-CH (residues 1-200 of human EB3)[30] solution were performed, incubating the grid for 30 s in between applications, and immediately freezing after the second application. Grids were stored in liquid nitrogen for no longer than a week before data acquisition.

**Data acquisition and processing.** Datasets were collected at the Joint Electron Microscopy Center at ALBA (JEMCA) at the ALBA synchrotron (Barcelona, Spain). A 200 kV Glacios TEM equipped with extreme field emission gun (X-FEG) optics, a cryogenic sample manipulator robot for up to 12 grids, and with a Falcon 4 direct electron detector (Thermo Fisher Scientific) was employed.

The data were processed with Relion 3.1.2 (Supplementary Table 3)[73,74]. Briefly, the movies were drift-corrected and dose-weighted using MotionCor2[75]. The motion-corrected micrographs were used to estimate the contrast transfer function parameters using Gctf[76]. The microtubules were picked manually or automatically using an undecorated microtubule model containing 14 protofilaments as references. The particles were extracted with a box size of 664 pixels, re-scaled to 150 pixels, and averaged to superparticles. The super-particles were classified into different numbers of protofilaments (Supplementary Fig. 6E). The 13-protofilament microtubule particles of each dataset were used for processing as previously described[45] with adapted scripts written in-house. Briefly, the extracted particles were used to generate an initial 3D model. This model was used as a reference for a subsequent supervised particle classification, to determine the tubulin register and seam location. Selected particles were subjected to 3D refinement. The angular distribution of particles demonstrated uniform and sufficient coverage across all orientations around the microtubule cross-section, supporting a reliable 3D reconstruction (Supplementary Fig. 6H). The final maps were unbinned with a box size of 600 pixels. Two maps of each dataset were obtained (Supplementary Table 3). The first map was produced without symmetry (C1) and the second map was obtained with helical symmetrization (sym) taking the position of the seam into account. The C1 maps have a resolution between 3.8 and 4.3 Å and were used to determine the protofilament twist and dimer rise with the relion_he-lix_toolbox command. To determine the errors of lattice parameters for each condition, each particle dataset was split into three smaller

subsets. Briefly, microtubules were first arranged by length and then evenly distributed across three subsets for processing in RELION separately. As a result, each subset contained a similar number of microtubules with comparable lengths and uniform particle angle distribution (Supplementary Fig. 6H), ensuring a consistent total particle count across the datasets. These subsets were processed as above described for whole datasets. The reported errors correspond to the standard deviation of the lattice parameter values obtained from these three subsets. The symmetrized maps reached a higher resolution (between 3.6 and 3.7 Å) and have been deposited at EMDB data bank with the following accession numbers: EMD-50172 (undecorated 13-protofilament αE254Q-microtubules), EMD-50174 (undecorated 13-protofilament 20%Q + N-microtubules), EMD-50177 (EB3-decorated 13-protofilament αE254Q-microtubules) and EMD-50178 (EB3-decorated 13-protofilament 20%Q + N-microtubules).

**Atomic model building and refinement.** Atomic models of undecorated and EB3-CH-decorated αE254Q- and 20%Q + N-microtubules were built in *Coot*[77] using the symmetrized maps. Initially, a tubulin dimer model obtained from recombinant WT-tubulin (PDB: 7SJ7)[32] was rigid-body fitted into the electron density of our symmetrized maps using the jiggle-fit tool. Subsequently, the αE254Q residue was mutated, and GDP at the E-site was replaced by GTP. The fitting of the protein chains into electron densities was conducted using the morph-fit and chain-fit tools, previous generation of self-restrains at 4.3 to prevent the deformation of secondary structures during the fitting within the maps. Geometry and secondary structure outliers of the initial dimer were manually adjusted, and the dimer model was refined in Phenix using the phenix.real_space_refine program[78,79]. The refined dimer was propagated within the symmetrized map by rigid-body fitting using the jiggle-fit tool in *Coot* to generate a three protofilament model each containing a central tubulin dimer flanked by one β-tubulin chain above and one α-tubulin chain below. One round of refinement was performed to build final models (Supplementary Table 3). Example regions showing the quality of symmetrized electron density maps (Supplementary Fig. 7) were generated using UCSF ChimeraX[80]. Final models of symmetrized maps have been deposited at the Protein Data Bank (PDB) with the following accession numbers: 9F3B (undecorated 13-protofilament αE254Q-microtubules), 9F3H (undecorated 13-protofilament 20%Q + N-microtubules), 9F3R (EB3-decorated 13-protofilament αE254Q-microtubules) and, 9F3S (EB3-decorated 13-protofilament 20%Q + N-microtubules).

C1 atomic models were built by rigid-body fitting of the refined models obtained from the symmetrized maps into respective C1 maps using the jiggle-fit tool in *Coot* without any further structure refinement. Structural representations (Fig. 5B–E and Supplementary Fig. 8) were generated from the C1 models using PyMOL (The PyMOL Molecular Graphics System, Version 1.2r3pre, Schrödinger, LLC).

**Reporting summary**
Further information on research design is available in the Nature Portfolio Reporting Summary linked to this article.

# Data availability
The authors declare that the data supporting the findings of this study are available within the paper, its supplementary information files and in the provided source data file. The atomic model of undecorated GDP microtubules with PDB accession number 7SJ7, described in ref. 32 was employed in this manuscript. Atomic coordinates of the symmetrized models have been deposited in the Protein Data Bank (PDB) with accession numbers 9F3B (undecorated 13-protofilament αE254Q-microtubules), 9F3H (undecorated 13-protofilament 20%Q + N-microtubules), 9F3R (EB3-decorated 13-protofilament αE254Q-microtubules) and, 9F3S (EB3-decorated 13-protofilament 20%Q + N-microtubules). The corresponding cryo-EM density maps have been deposited in the Electron

Microscopy Data Bank under the accession numbers EMD-50172 (undecorated 13-protofilament αE254Q-microtubules), EMD-50174 (undecorated 13-protofilament 20%Q + N-microtubules), EMD-50177 (EB3-decorated 13-protofilament αE254Q-microtubules) and EMD-50178 (EB3-decorated 13-protofilament 20%Q + N-microtubules). The mass spectrometry proteomics data have been deposited to the ProteomeXchange Consortium via the PRIDE[71] partner repository with the dataset identifier PXD058083. Source data are provided with this paper.

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

## Acknowledgements

The authors thank Pablo Guerra from the IBMB-CSIC Cryo-EM Platform at ALBA for support for cryo-EM sample preparation and data acquisition, Guadalupe Espadas-García and Eduard Sabido-Aguade from the CRG/UPF Proteomics Unit for data analysis, support in experimental design and sample preparation for the LC/MS-MS experiments, Andrea E. Prota for technical support for microtubule atomic model building and Eva Nogales, UC Berkeley, for helpful discussions. T.S. acknowledges the support of the Spanish Ministry of Science and Innovation through the Centro de Excelencia Severo Ochoa (CEX2020-001049-S, MCIN/AEI/

10.13039/501100011033), the Generalitat de Catalunya through the CERCA programme and to the EMBL partnership. T.S. also acknowledges funding from the European Research Council (ERC) under the European Union's Horizon 2020 research and innovation programme (grant agreement No 951430) and from the Spanish Ministry of Science and Innovation (grant PID2019-108415GB- I00/AEI/10.13039/501100011033). J.E-G was supported by the Spanish Ministry of Science and Innovation, "Juan de la Cierva—Formación" grant (FJC2020-043857-I). M.O.S. acknowledges financial support from the Swiss National Science Foundation (310030_192566). The Cryo-EM Platform at ALBA was supported by the Generalitat de Catalunya (project IU16-014045 (CRYO-TEM)) by the European Union (project "ERDF A way of making Europe"). The proteomics analyses were performed in the CRG/UPF Proteomics Unit which is part of the Spanish National Infrastructure for Omics Technologies (ICTS OmicsTech).

## Author contributions

J.E.-G., M.O.S., and T.S. conceptualized and designed the experimental research. J.E.-G., T.B.B., and F.R. performed the experiments and analyzed the data. J.E.-G. and F.R. performed the setup of TIRF and IRM microscopy. R.G.-C. generated protein constructs. J.E.-G. and M.G. expressed proteins. J.E.-G., S.S., and R.G.-C. set up and performed protein purification. J.E.-G., M.O.S., and T.S. wrote the manuscript.

## Competing interests

The authors declare no competing interests.
