## [Peer review File · Nature Communications]

REVIEWER COMMENTS

Reviewer #1 (Remarks to the Author):

This manuscript by Estévez-Gallego concerns the structure of the "GTP cap", arguably the most important region of the microtubule lattice. There is a long history of using nucleotide analogues of GTP to mimic the GTP cap, starting originally with the use of GMPCPP in the early 1990's. Mimics of the cap have been critical in explaining the mechanisms of EB family proteins and dynamic instability per se. More recently, the Surrey lab and others have developed new mimics using hydrolysis-deficient mutants of recombinant tubulin. The data presented here makes an interesting contribution to this research area, using a mixture of tubulins to create the most "faithful" mimic of the GTP cap to date.

The results are surprising, in that it's a heterogenous lattice that produces the "best results", more specifically an 80/20 mix of two hydrolysis-mutants, as though microtubule lattices follow the Pareto principle. More specifically, the mixed lattices grow like WT microtubules, nucleate like WT microtubules, have similar persistence lengths, and form mostly 13 pf microtubules. They solve the cryo-EM structure of these mixed lattices and compare the lattice parameters with those of other mimics, mutants, both w/ and w/out EB proteins.

Although I liked the paper, the presentation of results needs substantial improvement, and the standards for data presentation are below similar papers in this field. Perhaps most surprisingly, no cryo-EM structure of the "faithful" lattice is displayed, only a single cryo-EM image (Fig. 5B), so it's a structure paper that does not show the structure. The final resolution is stated at 3.6 Å so I'm sure it's a beautiful reconstruction!

-- Further, I would expect to see overlapped structures with the GDP lattice or some other visualization of conformational changes. If I understand correctly, the difference from the earlier Nogales comparisons will be shorter compaction and less change in twist. I would also be interested in the conformational changes between the EB-bound structure and the undecorated structure. Would that tell us something about what happens when EB dissociates?

-- As another example where improvement is needed, consider how growth rates are presented. Table 1 shows growth rates on a scale of ? to +++, which is hardly quantitative, especially compared to Table 2, which provides the on-rate constants extracted from data in Fig. 4B. I dislike +/- scales in general, and these two tables demonstrate why: because converting the +'s into growth rates gives unequal values. In addition, the values in Table 2 don't make sense to me. Assuming the apparent equation for growth is $k_{on} \times \text{Tubulin} - k_{off}$, then the y-intercept should be negative (and the C_c as

defined by Oosawa should be positive). That's not the case in Fig. 4B--the y-intercepts of the lines are positive! Thus the k_{off} 's in Table 2 should be negative, so what's going on? Figure 4B also contains the same data as presented in Suppl. Fig. 2 (namely growth at 10 μ M tubulin)---so why the separate figure and presentation?

-- Figure 1 is currently only a schematic, which could be improved and placed alongside the actual data on twist and compaction, rather than as a separate figure, e.g., in a revamped structure figure.

-- Some data in the supplement seems like it could be moved to main figures, e.g., the L_p measurements in S3D. The L_p measurements get a full paragraph in the results, after all.

-- Figure 3 shows the results of a titration of E254Q. Oddly, the data for the "faithful" ratio, 20Q+N is not plotted on Fig. 3B, because there is no data point where the dashed line intersects the fit. Yet the kymograph data at 20Q+N is shown in Fig. 3C. I assume this is a glitch.

Overall, the data are strong and interesting, but the manuscript needs significant improvement.

Reviewer #2 (Remarks to the Author):

Estevez-Gallego et al. employ cryo-EM, TIRF microscopy, and recombinant tubulin to identify a mixture of two hydrolysis-deficient α -tubulin mutations that could mimic the polymerization rates, the end-binding protein decoration pattern, and the structure of GTP-bound microtubule lattice (i.e., the GTP cap). This study provides a novel approach for dissecting the molecular mechanism underlying the dynamic instability of microtubules. This work is suitable for publication in Nature Communications after addressing some concerns.

Concerns:

1. The current manuscript is a bit short of understanding the mechanism at the molecular level. It would be better if the authors could reveal or provide a possible model from their structural data to explain why even a 1 Da change in E254Q could cause changes in microtubule properties and structures. For example, what structural elements lead to curved microtubules when E254Q is copolymerized with EB3 in the presence of GTP? This information could help understand the role of the GTP cap in microtubule dynamic instability as the tubulin undergoes curved to straight conformational changes during the lattice incorporation.

2. What is the ratio of E254Q and E254N that are being incorporated into the lattice of mosaic microtubules? By providing this information on the contribution of each construct in the microtubule lattice, this paper could provide essential parameters for future in silico simulations of microtubule dynamic instability.
3. Is there any statistical analysis regarding the measured twist and rise values of different microtubules? As the resolution of the C1 symmetry is $\sim 4 \text{ \AA}$, it will be critical to address the significance of their observation.
4. How did the author generate the labeled tubulin to polymerize microtubules in their TIRF assays? There needs to be more information on this aspect in the methods.
5. The label of 20%Q+N in Figure 4E appears not to be a 'diamond.' The author needs to revise the figure/figure legend.

Reviewer #3 (Remarks to the Author):

The microtubule is a dynamically unstable polymer with its plus end exhibiting periods of stochastic growth and shrinkage. The GTP cap stabilizes the microtubule plus end, and erosion of the cap results in microtubule catastrophe. Slowly hydrolysable GTP analogues and tubulin mutations that abolish GTP hydrolysis have been studied extensively to understand the nature of the GTP cap, however the true structure of the cap remains poorly understood. To answer this question Estévez-Gallego, J. and colleagues generated a mutation in the α -tubulin that abolished GTP hydrolysis (α E254Q), a commonly inactivating mutation used in GTPases. The authors discover that α E254Q when mixed in a specific ratio with another hydrolysis-mutant tubulin (α E254N) mimics the GTP cap most closely. This is a cute result, but hardly surprising based on work on the nature of the cap already out there regarding mixed GTP and GDP states. Moreover, the manuscript lacks proper statistics throughout, some directly related to the robustness and reproducibility of the experiments presented. Several key pieces of data discussed are not shown anywhere in the manuscript. Lastly, the writing and organization of the manuscript also requires more work. While it has promise, in its current state this work is not publishable in any venue.

Major concerns:

Figure 1

Figure 1 is an attempt to display basic microtubule properties. In what way does fig. 1A show GTP dimers adding to the growing (+) end. What are the colors, and what does the shading gradient represent? More information is needed in the legend. Perhaps the authors should look at some reviews (of which there are many), with a similar diagram but with much better explanation of the figure. Panel B is misleading; it suggests that the lattice expansion results in a larger α -tubulin subunit. To correct the figure, the authors should draw tubulin subunits of the same size, and show less extensive (weaker) microtubule longitudinal interface in the expanded lattice. The colors chosen for the schematics are very misleading. The colors imply that the compacted longitudinal interface in blue-gray dimer forms straight no twist microtubules (shown below), while the yellow-green dimer with the expanded longitudinal interface forms twisted microtubule lattices, which is not true. The figures in a manuscript should show novel findings. This figure could either be a supplementary figure or removed completely and the reader can be directed to one of numerous reviews describing the microtubule plus end structure and dynamic properties as well as to papers by Dennis Chretien describing microtubule lattice parameters.

Figure 2:

Panel B should also include the E254A mutant for comparison. Panel C: if the error bars are SEM they are quite large on the WT. Perhaps this is to be expected due to differing amount of GTP tubulin, but there is no quantitative description of the data. i.e. how many times was this performed? If you show a bar graph with error bars, report the N so that the reader can get a sense of the repeatability. This is a major problem in this paper and occurs repeatedly. Panels D&E: These panels show the localization of EB, but it is very difficult to see the EB in the merged image. Please show the channels separately, in addition, a few linescans would be a nice way to show the difference in localization, and the difference in EB intensity. Why is the GFP signal in E not constant and displays periods of lower and higher intensity. The kymographs seem a little squished. They would be easier to understand if they were made taller. (F&G). The binding curve does not reach saturation, but was used to determine K_d . More datapoints are needed at higher concentrations to reliably measure the K_d . They state “n=100 microtubules” was that from one chamber? How many independent experiments did this data come from? These MTs don’t look excessively curved (as they state later). (H) I do not understand the point of this figure. Are they simply showing rare events? To what end? Is there mechanism or biochemical information from these rare events? In the text they state that these are “rare” events, but there is NO quantification of how frequently they observe them. They have the kymographs, so they need to quantify. How often is it observed, or is this a figure from a few weird MT out of a population of 100? The authors should include the movies in SI that these kymos came from so the reader has better understanding of the dynamic nature of these events. They also show end splitting in H5. How often does this occur? Does the fact that EB recognizes the curled tip and the straight tip yield any significant insight about EB binding? Again, this would be easier to interpret if the channels were displayed separately and it was also presented as a movie. The text describing Fig.2 uses adjectives like “occasional” and “more frequently”, which make their observations highly subjective. These are all quantifiable parameters.

Figure 3: In the text they describe “segmented” EB binding to the E2N mutant. I do not see this in the figure or in the chosen kymographs they present. Perhaps some of the MTs in the E2N display a segmented EB signal but it is not very clear. They should highlight these MTs and/or show a gallery of line scans in SI. I understand this was shown in another paper by the authors, but it is not clearly presented in this figure. Include line scans of several MTs in each condition. There is no quantification of this provided. I see patches of light and dark EB signal in all the kymographs, is this “segmented”? If they mean regions with no binding, I don’t see that on any of the kymographs presented. Line scans would also give the reader an idea about the relative brightness of the EB on the lattice in each condition. The statement “demonstrating the ability of E2N mutants to adopt extended stretched of different lattice conformations was abolished” should be removed. There is no quantification provided for this, and it is not at all clear from the images. This entire figure could also be shown as movie files to give the reader a better understanding of the dynamic differences between titrations. A missing control is the titration also with WT tubulin. The E2N and E2Q have substantially different growth rates (0% and 100% in 3B) does the fact that this change is nonlinear tell you anything about the tubulin incorporation (is the E2Q forming ‘islands’ or being evenly distributed ?) There is also a major weakness with these titration experiments: we have no idea how the different mutants incorporate into the lattice and that has a direct impact on the interpretation of the experiments. Figure 3B is an exponential growth to plateau fit. Why did the authors choose this model and not others, and what does the fit parameter represent? If the goal is to find the equivalent mixture to WT growth rates, they should put figure S2 in the main figure.

Figure 4: Please state how many independent experiments were performed. This is a problem throughout the manuscript which raises serious questions about the rigor and robustness of this work. The number of microtubules is not sufficient, since they can come from just one experiment. The legend states “ WT- (circles, green line), 964 α E254Q- (triangles, blue line), α E254N- (squares, orange line), and 20%Q+N-microtubules 965 (diamonds, purple line).” But there are no triangles on the blue line, or diamonds on the purple line. This is an important figure and needs to be corrected. Why were all the mutants and WT not assayed at the same concentrations? Are these data mean with SEM again, or is this SD?

Figure 5. In E, please provide the WT lattice protofilament numbers.

The last section in the RESULTS describes cryo-EM structural findings. Where are the structures? Table S3 lists four high-resolution cryo-EM reconstructions, but there is not a single figure showing these cryo-EM maps. Are there any atomic models derived from these maps? The authors need to have one main figure showing cryo-EM reconstructions and atomic models, another figure should have atomic models aligned to each other to show lattice geometry variations described on the page 9 of the manuscript. Additionally, there needs to be at least one figure and detailed

description about comparison of the microtubule lateral and longitudinal interfaces between WT and E254Q- and 269 20%Q+N-microtubules, so that a reader understands how different microtubule lattices described in this manuscript are assembled. Finally, the authors need to add a supplementary figure detailing cryo-EM image processing and a supplementary table listing atomic models statistics. This has been the standard in the field for years.

Page 8. “Mammalian GDP-microtubules polymerized from WT-tubulin in the presence of GTP are known to exhibit a Gaussian distribution of various protofilament numbers centered around an average of 13 protofilaments”. Dynamic undecorated mammalian microtubules in vitro form 14 protofilament microtubules as the major class (Zhang, R. et. al., PNAS 2018). Therefore, the statement “whereas 20%Q+N-microtubules were predominantly composed of 13 protofilaments, adding yet another piece of evidence that 20%Q+N-282 microtubules eliminate mutation-specific effects and faithfully resemble WT GTP- microtubules” has no merit behind it.

Figure 5 Panel B. The panels shows that 20%Q+N forms both microtubules and sheets. The second microtubule from the left has two lattice types. The upper half of this microtubule lattice looks like that in the panel A (α E254Q) and the lower half of this microtubule looks like the first microtubule from the left in the panel B. Given the visible differences in the lattice geometry per microtubule and the mixture of sheets and tubes, how did the authors manage to obtain a high-resolution reconstruction? Were they able to separate different lattices present in 20%Q+N microtubule before obtaining the final structure? Or is the final structure representing an average of the two lattices? The manuscript lacks a detailed description of the cryo-EM image processing, a minimum for publication in any journal. No cryo-EM maps are shown, no information about data processing, so impossible to assess the validity of much described here.

Did the authors exclude 20%Q+N sheets from the protofilament number quantification?

Panel F. Please describe how exactly the dimer rise and protofilament twist were obtained.

Other issues:

- Page 4. Figure referenced should be Fig. 2E in “This was in contrast to 139 dynamically growing and shrinking WT microtubules displaying EB3 comets only at their 140 growing ends (Fig. 1E,..”
- The sentence on p9 “20%Q+N microtubules show dimer rise more similar to that...” requires citations

- Figure S3D and table 2. They calculate the persistence length via a curvature distribution. Their values seem quite low, comparable to taxol treated brain MT. They should give the length distribution of the polymers used. Were they calculated at the same growth rate? The fact that the mutant mix is similar to the WT (actually a little larger). Does this imply that lattice strain is a driving factor in the persistence length measurements?

Point-by-point response to the reviewers' comments

Reviewer #1 (Remarks to the Author):

This manuscript by Estévez-Gallego concerns the structure of the "GTP cap", arguably the most important region of the microtubule lattice. There is a long history of using nucleotide analogues of GTP to mimic the GTP cap, starting originally with the use of GMPCPP in the early 1990's. Mimics of the cap have been critical in explaining the mechanisms of EB family proteins and dynamic instability per se. More recently, the Surrey lab and others have developed new mimics using hydrolysis-deficient mutants of recombinant tubulin. The data presented here makes an interesting contribution to this research area, using a mixture of tubulins to create the most "faithful" mimic of the GTP cap to date.

The results are surprising, in that it's a heterogenous lattice that produces the "best results", more specifically an 80/20 mix of two hydrolysis-mutants, as though microtubule lattices follow the Pareto principle. More specifically, the mixed lattices grow like WT microtubules, nucleate like WT microtubules, have similar persistence lengths, and form mostly 13 pf microtubules. They solve the cryo-EM structure of these mixed lattices and compare the lattice parameters with those of other mimics, mutants, both w/ and w/out EB proteins.

Although I liked the paper, the presentation of results needs substantial improvement, and the standards for data presentation are below similar papers in this field. Perhaps most surprisingly, no cryo-EM structure of the "faithful" lattice is displayed, only a single cryo-EM image (Fig. 5B), so it's a structure paper that does not show the structure. The final resolution is stated at 3.6 Å so I'm sure it's a beautiful reconstruction!

We thank the reviewer for this overall positive evaluation. We have improved the presentation, as requested, and generated the missing reconstructions, as explained in detail in our point-by-point response below.

-- Further, I would expect to see overlapped structures with the GDP lattice or some other visualization of conformational changes. If I understand correctly, the difference from the earlier Nogales comparisons will be shorter compaction and less change in twist. I would also be interested in the conformational changes between the EB-bound structure and the undecorated structure. Would that tell us something about what happens when EB dissociates?

We have added the requested structures derived from the C1 maps of pure α E254Q microtubules, 20%Q+N mosaic microtubules, and the two EB3-decorated microtubules. The new data are presented in the mostly new Fig. 5, in the new Suppl. Figs 6 and 7 and in the new Table S4. We present the requested overlays in Fig. 5D, E and Suppl. Fig. 7, showing changes upon nucleotide change and EB3 binding for both pure α E254Q microtubules and 20%Q+N mosaic microtubules.

-- As another example where improvement is needed, consider how growth rates are presented. Table 1 shows growth rates on a scale of ? to +++, which is hardly quantitative, especially compared to Table 2, which provides the on-rate constants extracted from data in Fig. 4B. I dislike +/- scales in general, and these two tables demonstrate why: because converting the +'s into growth rates gives unequal values.

Table 1 is intended to give an overview of previous published observations with wild type and mutant recombinant human microtubules. Not all of these observations were quantified in the existing literature, but the available data nevertheless allows a first qualitative comparison with the new E254Q microtubules. In our view, the table makes these comparisons more accessible for the reader who is not familiar with the previous work. We have now added the relevant literature citations to Table 1 to make its purpose clear and start calling it already in the Introduction. In Table 2, we moved on to comparing the 20%Q+N mosaic microtubules with pure Q, pure N, and wild type microtubules. Since these comparisons address the central point of the manuscript, we made sure that these comparisons are all carefully quantified.

In addition, the values in Table 2 don't make sense to me. Assuming the apparent equation for growth is $k_{on} \times \text{Tubulin} - k_{off}$, then the y-intercept should be negative (and the C_c as defined by Oosawa should be positive). That's not the case in Fig. 4B--the y-intercepts of the lines are positive! Thus the k_{off} 's in Table 2 should be negative, so what's going on?

We apologize for having chosen an unfortunate display of the graph in Fig. 4B - the x-axis did not go through $y = 0$. We now show the x-axis going through $y=0$, which makes it more evident that the regressions cut the y-axis at negative values, meaning that all k_{off} values are positive, as expected and in agreement with the values reported in Table 2.

Figure 4B also contains the same data as presented in Suppl. Fig. 2 (namely growth at 10 μM tubulin)--so why the separate figure and presentation?

This point addresses the style of presentation. Our intention is to first present a simpler dataset, comparing speeds at a single tubulin concentration, for which speeds for all four types of microtubules could be measured (Suppl. Fig. 3). We then present a more complex dataset, including the dependence on the tubulin concentration; to show that 20%Q+N microtubules grow like wild type microtubules over the entire range of tubulin concentrations tested. We have now added a statement to the legend in Suppl. Fig. 3, making it clear that these are the same data as shown in Fig. 4B for the 10 μM condition, but plotted in a different style, for clarity.

-- Figure 1 is currently only a schematic, which could be improved and placed alongside the actual data on twist and compaction, rather than as a separate figure, e.g., in a revamped structure figure.

We feel that for a non-specialist, this schematic figure is useful, to illustrate the concepts introduced in the Introduction section. We made it more compact so that its size can be reduced in the final version of the paper. We also improved the schematics, as requested by Reviewer 3.

-- Some data in the supplement seems like it could be moved to main figures, e.g., the Lp measurements in S3D. The Lp measurements get a full paragraph in the results, after all.

We kept the paragraph in the main text about the persistence length measurements concise and prefer to show these data in the Supplement. We improved however the clarity of the corresponding Methods section.

-- Figure 3 shows the results of a titration of E254Q. Oddly, the data for the "faithful" ratio, 20Q+N is not plotted on Fig. 3B, because there is no data point where the dashed line intersects the fit. Yet the kymograph data at 20Q+N is shown in Fig. 3C. I assume this is a glitch.

We show in Fig. 3B how we measured the dependence of the growth speed on the composition of the mosaic microtubules. This measurement was then used to find the mixing ratio at which mosaic microtubules grow with wild type speed. That happened to be at a ratio that was not tested in the initial exploration. We then confirmed that at 20% E254Q tubulin in the mosaic microtubules (images in Fig. 3C), the growth speed is indeed as for wild type microtubules (Fig. 4B, Suppl. Fig. 3). We believe that this is a logical way to present the data.

Overall, the data are strong and interesting, but the manuscript needs significant improvement.

We thank this reviewer for their constructive criticism and feel that our changes throughout the manuscript have improved the presentation.

Reviewer #2 (Remarks to the Author):

Dear Editors,

Estevez-Gallego et al. employ cryo-EM, TIRF microscopy, and recombinant tubulin to identify a mixture of two hydrolysis-deficient α -tubulin mutations that could mimic the polymerization rates, the end-binding protein decoration pattern, and the structure of GTP-bound microtubule lattice (i.e., the GTP cap). This study provides a novel approach for dissecting the molecular mechanism underlying the dynamic instability of microtubules. This work is suitable for publication in Nature Communications after addressing some concerns.

We thank this reviewer for their overall positive evaluation.

Concerns:

1. The current manuscript is a bit short of understanding the mechanism at the molecular level. It

would be better if the authors could reveal or provide a possible model from their structural data to explain why even a 1 Da change in E254Q could cause changes in microtubule properties and structures. For example, what structural elements lead to curved microtubules when E254Q is copolymerized with EB3 in the presence of GTP? This information could help understand the role of the GTP cap in microtubule dynamic instability as the tubulin undergoes curved to straight conformational changes during the lattice incorporation.

We agree with the reviewer that it is an interesting question how such a small change can induce the growth of curved microtubules. Whether this curving of closed E254Q mutant tubes is directly related to the curvature of sheets formed at growing microtubule ends before they close into a tube is unknown. In our view, it is not clear, why the different tubulin mutations and the different non-hydrolyzable GTP analogs all cause different characteristic abnormalities. We are unaware of a model that could explain these deviations from normal behavior and we believe that answering this is beyond the scope of the current study. Instead, our focus is on finding out how one can overcome such generation of abnormalities. Our detailed biochemical characterizations show that, quite remarkably, as we find, the 20%Q+N mosaic microtubules do not display such abnormalities anymore, likely providing now a faithful mimic of the GTP state of a microtubule.

2. What is the ratio of E254Q and E254N that are being incorporated into the lattice of mosaic microtubules? By providing this information on the contribution of each construct in the microtubule lattice, this paper could provide essential parameters for future in silico simulations of microtubule dynamic instability.

We agree that this is an important point. We have therefore performed additional mass spectrometry analysis on mosaic microtubules. These new data demonstrate that 20% E254Q tubulin is indeed incorporated in the microtubules that polymerize in a mixture of 20% E254Q and 80% E254N tubulin. These new data are now shown in Suppl. Fig. 4, and are presented in the main text on lines 202-208, and the method is described in the Methods section (lines 621-645).

3. Is there any statistical analysis regarding the measured twist and rise values of different microtubules? As the resolution of the C1 symmetry is $\sim 4 \text{ \AA}$, it will be critical to address the significance of their observation.

We considered splitting our datasets into two parts to generate separate maps for comparison, but given the size of our datasets this was not considered prudent. We note that such statistical analysis is currently not the standard in the field, probably because different labs demonstrated that the global lattice parameters can be determined reliably and consistently for different microtubule lattices from reconstructions with similar resolution as ours.

4. How did the author generate the labeled tubulin to polymerize microtubules in their TIRF assays? There needs to be more information on this aspect in the methods.

Our recombinant tubulin is unlabeled and growing microtubules are therefore either observed using the label-free microscopy method IRM, or using TIRF microscopy by visualizing GTP-deficient microtubules or growing wild type microtubule ends indirectly via GFP-tagged EB3. Fluorescent CF640R-tubulin is used only to visualize the GMPCPP-stabilized and surface-immobilized microtubule seeds from which the recombinant microtubules grow. We rearranged and expanded the corresponding Legends and Methods sections to make this clearer.

5. The label of 20%Q+N in Figure 4E appears not to be a 'diamond.' The author needs to revise the figure/figure legend.

We thank for spotting this typo in the legend, which has now been corrected.

Reviewer #3 (Remarks to the Author):

The microtubule is a dynamically unstable polymer with its plus end exhibiting periods of stochastic growth and shrinkage. The GTP cap stabilizes the microtubule plus end, and erosion of the cap results in microtubule catastrophe. Slowly hydrolysable GTP analogues and tubulin mutations that abolish GTP hydrolysis have been studied extensively to understand the nature of the GTP cap, however the true structure of the cap remains poorly understood. To answer this question Estévez-Gallego, J. and colleagues generated a mutation in the α -tubulin that abolished GTP hydrolysis (α E254Q), a commonly inactivating mutation used in GTPases. The authors discover that α E254Q when mixed in a specific ratio with another hydrolysis-mutant tubulin (α E254N) mimics the GTP cap most closely. This is a cute result, but hardly surprising based on work on the nature of the cap already out there regarding mixed GTP and GDP states.

We are unfortunately unsure which literature on mixed GDP and GTP states the reviewer refers to. We do therefore not see how previous work could predict that mixing two GTPase deficient mutants at a certain ratio generates GTP-containing microtubules with biochemical properties that are remarkably similar to growing GTP ends of wild type microtubules - in contrast to microtubules polymerized only from one type of mutant or in the presence of a non-hydrolyzable GTP analog. We therefore believe that the novelty of our work presented here is not compromised by previous work.

Moreover, the manuscript lacks proper statistics throughout, some directly related to the robustness and reproducibility of the experiments presented. Several key pieces of data discussed are not shown anywhere in the manuscript. Lastly, the writing and organization of the manuscript also requires more work. While it has promise, in its current state this work is not publishable in any venue.

We improved these aspects of the presentation of our study, as detailed in our point-by-point reply below.

Major concerns:

Figure 1

Figure 1 is an attempt to display basic microtubule properties. In what way does fig. 1A show GTP dimers adding to the growing (+) end. What are the colors, and what does the shading gradient represent? More information is needed in the legend. Perhaps the authors should look at some reviews (of which there are many), with a similar diagram but with much better explanation of the figure. Panel B is misleading; it suggests that the lattice expansion results in a larger α -tubulin subunit. To correct the figure, the authors should draw tubulin subunits of the same size, and show less extensive (weaker) microtubule longitudinal interface in the expanded lattice. The colors chosen for the schematics are very misleading. The colors imply that the compacted longitudinal interface in blue-gray dimer forms straight no twist microtubules (shown below), while the yellow-green dimer with the expanded longitudinal interface forms twisted microtubule lattices, which is not true. The figures in a manuscript should show novel findings. This figure could either be a supplementary figure or removed completely and the reader can be directed to one of numerous reviews describing the microtubule plus end structure and dynamic properties as well as to papers by Dennis Chretien describing microtubule lattice parameters.

We thank the reviewer for the suggestions to improve this figure. We agree that the schematics needed improvements and that some explanations were lacking in the legend. As suggested, we consulted similar schemes in the literature, and have now improved the figure taking the reviewer's suggestions on board. We also made the figure more compact so that it can take less space in the final version of the paper. We think that in its revised form, the figure is useful for the general reader to illustrate the lattice parameters that we introduce in the Introduction section. If deemed necessary, we can move the figure to the Supplement, but think that it makes the manuscript more accessible when presented as main figure.

Figure 2:

Panel B should also include the E254A mutant for comparison.

We added the requested E254A mutant data. They agree with the cited published data.

Panel C: if the error bars are SEM they are quite large on the WT. Perhaps this is to be expected due to differing amount of GTP tubulin, but there is no quantitative description of the data. i.e. how many times was this performed? If you show a bar graph with error bars, report the N so that the reader can get a sense of the repeatability. This is a major problem in this paper and occurs repeatedly.

We added the requested statistical information to the legend. We note that despite the error for the wild type condition being a bit larger, the result agrees with the previous measurement for such wild type microtubules (Roostalu et al., Elife, 2020).

Panels D&E: These panels show the localization of EB, but it is very difficult to see the EB in the merged image. Please show the channels separately, in addition, a few linescans would be a nice way to show the difference in localization, and the difference in EB intensity. Why is the GFP signal in E not constant and displays periods of lower and higher intensity. The kymographs seem a little squished. They would be easier to understand if they were made taller.

Compared to dual channel TIRF microscopy movies a major part of the fluorescence signal is lost in our combined IRM/TIRFM set-up. That's a technical limitation of such a combined set-up and the reason why the EB3 signal at wild type microtubule ends appears less bright. To clearly show the growing end localization at wild type ends, we chose to display the kymographs in Fig. 2E. The behavior of the wild type microtubules agrees with what has been observed previously for these microtubules. The main point here is to show the very different EB3 binding to wild type compared to E254Q microtubules, which is convincingly demonstrated by the data as presented. The "compressed" appearance of the kymographs is again a consequence of the combined IRM/TIRFM setup that results in a reduced imaging frequency of 1/15 s when imaging both modalities in an alternating manner. This does however not affect the message the kymographs convey, because the scale bars indicate the time and spatial scales.

As the reviewer noted, there is also some variation of the EB3 binding intensity of microtubules containing higher percentages of α E254Q tubulin or even in pure E254Q microtubules, but much less so than in α E254N microtubules. We have added a careful quantitative analysis of fluorescence intensities to stress this point, which we now show in the new Suppl. Fig. 2. We have also made the display of the fluorescence intensities in Figs. 3 and 4 more consistent and added details regarding fluorescence intensity analysis and display to the Methods section (lines 563-578).

(F&G). The binding curve does not reach saturation, but was used to determine K_d . More datapoints are needed at higher concentrations to reliably measure the K_d . They state "n=100 microtubules" was that from one chamber? How many independent experiments did this data come from? These MTs don't look excessively curved (as they state later).

We now show the binding curve using a linear x-axis (in contrast to logarithmic as before), which better allows to appreciate that binding saturates. The requested information about number of experiments has been added to the legend. The curved nature of the microtubules is not so obvious here because the microtubules are relatively short and were not co-polymerized with EB3, as we state more clearly now.

(H) I do not understand the point of this figure. Are they simply showing rare events? To what end? Is there mechanism or biochemical information from these rare events? In the text they state that

these are “rare” events, but there is NO quantification of how frequently they observe them. They have the kymographs, so they need to quantify. How often is it observed, or is this a figure from a few weird MT out of a population of 100? The authors should include the movies in SI that these kymos came from so the reader has better understanding of the dynamic nature of these events. They also show end splitting in H5. How often does this occur? Does the fact that EB recognizes the curled tip and the straight tip yield any significant insight about EB binding? Again, this would be easier to interpret if the channels were displayed separately and it was also presented as a movie. The text describing Fig.2 uses adjectives like “occasional” and “more frequently”, which make their observations highly subjective. These are all quantifiable parameters.

We agree with the reviewer that these rare events deviate the attention of the reader from the main points of the study, which are not based on rare events. To simplify the presentation and avoid confusion, we removed the presentation of the rare events, which were indeed so infrequent that they were hard to quantify reliably.

Figure 3: In the text they describe “segmented” EB binding to the E2N mutant. I do not see this in the figure or in the chosen kymographs they present. Perhaps some of the MTs in the E2N display a segmented EB signal but it is not very clear. They should highlight these MTs and/or show a gallery of line scans in SI. I understand this was shown in another paper by the authors, but it is not clearly presented in this figure. Include line scans of several MTs in each condition. There is no quantification of this provided. I see patches of light and dark EB signal in all the kymographs, is this “segmented”? If they mean regions with no binding, I don’t see that on any of the kymographs presented. Line scans would also give the reader an idea about the relative brightness of the EB on the lattice in each condition. The statement “demonstrating the ability of E2N mutants to adopt extended stretched of different lattice conformations was abolished” should be removed. There is no quantification provided for this, and it is not at all clear from the images. This entire figure could also be shown as movie files to give the reader a better understanding of the dynamic differences between titrations.

Segmented EB3 binding to E254N microtubules has indeed been demonstrated and quantitatively characterized in a previous publication (La France, PNAS, 2022). We added a quantitative analysis of the EB3 binding intensities, which confirms the segmented nature of EB3 binding to E254N microtubules in contrast to α E254Q or mosaic microtubules (see new Suppl. Fig. 2). As mentioned above, we also made the display of the intensities in the figures more consistent - see also Methods section (lines 563-578). This new quantification demonstrates that the fluctuations in EB3 binding affinity are strongly reduced with increasing fraction of the α E254Q mutant in the mixed α E254Q/N microtubule lattice.

A missing control is the titration also with WT tubulin.

The goal of this study was to generate a faithful GTP cap mimic, a microtubule that contains GTP throughout. Mixing experiments with wild type tubulin are therefore considered beyond scope, and will be part of a separate study, having a different goal.

The E2N and E2Q have substantially different growth rates (0% and 100% in 3B) does the fact that this change is nonlinear tell you anything about the tubulin incorporation (is the E2Q forming 'islands' or being evenly distributed ?)

We agree with the reviewer that this is an interesting observation. We now discuss in the Discussion section (lines 345-352) our interpretation of this result: growth speed is not only determined by the incoming soluble tubulin subunits, but likely also by the end structure of the growing microtubule, which by itself may depend on the mixing ratio.

There is also a major weakness with these titration experiments: we have no idea how the different mutants incorporate into the lattice and that has a direct impact on the interpretation of the experiments.

This point was also raised by Reviewer 2 and we agree that it is an important issue. We have now performed mass spectrometry experiments that confirm that the ratio of incorporated tubulin dimers reflects their ratio in solution. We added the new Suppl. Fig. 4 and a corresponding statement to the main text (lines 202-208) and the method is described in the Methods section (lines 621-645).

Figure 3B is an exponential growth to plateau fit. Why did the authors choose this model and not others, and what does the fit parameter represent? If the goal is to find the equivalent mixture to WT growth rates, they should put figure S2 in the main figure.

This is an empirical fit with the goal to find the ratio of mutants that results in wild type growth speed, as the reviewer states correctly. We prefer to leave the previous Fig. S2 (now Fig. S3) in the Supplement as it contains data already presented in a main figure, namely in Fig. 4B, as the reviewer noted earlier.

Figure 4: Please state how many independent experiments were performed. This is a problem throughout the manuscript which raises serious questions about the rigor and robustness of this work. The number of microtubules is not sufficient, since they can come from just one experiment. The legend states " WT- (circles, green line), 964 α E254Q- (triangles, blue line), α E254N- (squares, orange line), and 20%Q+N-microtubules 965 (diamonds, purple line)." But there are no triangles on the blue line, or diamonds on the purple line. This is an important figure and needs to be corrected. Why were all the mutants and WT not assayed at the same concentrations? Are these data mean with SEM again, or is this SD?

We agree that this is an important figure and apologize for the typos in the legend that have now been corrected - thanks for spotting. We have also provided the requested statistical information in the legend.

The different microtubule types were not assayed at the same concentration because they have different growth and nucleation properties, consistent with what is reported in other parts of the manuscript. Higher concentrations are needed for decent growth for E254N microtubules and E254Q microtubules tend to spontaneously nucleate at higher concentration causing tubulin depletion from the solution.

Figure 5. In E, please provide the WT lattice protofilament numbers.

We consider experiments with wild type microtubules beyond scope in the context of this study. Ti et al., *Dev Cell*, 2018 showed that recombinant human wild type microtubules of the same isotype composition as in our study here, display mostly 13 protofilaments. We cite this paper to refer the reader to this previous work.

The last section in the RESULTS describes cryo-EM structural findings. Where are the structures? Table S3 lists four high-resolution cryo-EM reconstructions, but there is not a single figure showing these cryo-EM maps. Are there any atomic models derived from these maps? The authors need to have one main figure showing cryo-EM reconstructions and atomic models, another figure should have atomic models aligned to each other to show lattice geometry variations described on the page 9 of the manuscript. Additionally, there needs to be at least one figure and detailed description about comparison of the microtubule lateral and longitudinal interfaces between WT and E254Q- and 269 20%Q+N-microtubules, so that a reader understands how different microtubule lattices described in this manuscript are assembled. Finally, the authors need to add a supplementary figure detailing cryo-EM image processing and a supplementary table listing atomic models statistics. This has been the standard in the field for years.

Despite having used a 200 keV microscope that is available to us in Barcelona (instead of a 300 keV microscope) which limits the quality of the maps, we were able to derive atomic models for undecorated and EB3-decorated α E254Q and 20%Q+N microtubules from our symmetrized maps. Detailed statistics regarding the model building process are now provided in Suppl. Table 4. Although direct atomic model building from C1 maps was not possible, we succeeded in capturing the structural differences in dimer rise and supertwist through rigid body fitting of the corresponding models derived from our symmetrized maps. This allowed us to show the differences in lattice geometry in the new Fig 5 and the new Suppl. Fig. 7, as requested.

Page 8. "Mammalian GDP-microtubules polymerized from WT-tubulin in the presence of GTP are known to exhibit a Gaussian distribution of various protofilament numbers centered around an average of 13 protofilaments". Dynamic undecorated mammalian microtubules in vitro form 14 protofilament microtubules as the major class (Zhang, R. et. al., *PNAS* 2018). Therefore, the

statement “whereas 20%Q+N-microtubules were predominantly composed of 13 protofilaments, adding yet another piece of evidence that 20%Q+N-282 microtubules eliminate mutation-specific effects and faithfully resemble WT GTP- microtubules” has no merit behind it.

It is well established that the protofilament number of microtubules is influenced by the buffer. When wild type (mammalian brain) microtubules are grown in the presence of high concentrations of glycerol, as in the publication cited by the reviewer (Zhang, R. et. al., PNAS 2018), an increased protofilament number is expected compared to microtubules not grown in glycerol whose protofilament number centers around 13 (Estévez-Gallego et al., Elife, 2020). In our cryo-EM experiments, we polymerize the recombinant microtubules in the absence of glycerol, in contrast to many other cryo-EM experiments (Ray et al., JCB, 1993; Alushin et al., Cell, 2014; Zhang et al., PNAS, 2018). Wild type microtubules grown from the same recombinant isotype combination as ours have previously been reported to have 13 protofilaments in the absence of glycerol (Ti et al., Dev Cell, 2018). The expectation of wild type microtubules preferentially displaying 13 protofilaments in the absence of glycerol is therefore valid.

Figure 5 Panel B. The panels shows that 20%Q+N forms both microtubules and sheets. The second microtubule from the left has two lattice types. The upper half of this microtubule lattice looks like that in the panel A (α E254Q) and the lower half of this microtubule looks like the first microtubule from the left in the panel B. Given the visible differences in the lattice geometry per microtubule and the mixture of sheets and tubes, how did the authors manage to obtain a high-resolution reconstruction? Were they able to separate different lattices present in 20%Q+N microtubule before obtaining the final structure? Or is the final structure representing an average of the two lattices? The manuscript lacks a detailed description of the cryo-EM image processing, a minimum for publication in any journal. No cryo-EM maps are shown, no information about data processing, so impossible to assess the validity of much described here.

Did the authors exclude 20%Q+N sheets from the protofilament number quantification?

We thank the reviewer for bringing this to our attention. Panels A and B were indeed mixed up. Apologies. We have rectified this error by presenting the corrected micrographs in the new Suppl. Fig. 6 (the previous image in panel A now corresponds to α E254Q, while panel B contains a micrograph of 20%Q+N microtubules).

In the process of generating our maps and atomic models, open structures were excluded and solely 13_3 microtubule particles were used for the generation of the atomic models that we now add to this revised version of the manuscript. We also expanded the description of the image processing, as requested.

Panel F. Please describe how exactly the dimer rise and protofilament twist were obtained.

These lattice parameters were obtained from the C1 maps with the `relion_helix_toolbox` command, as now mentioned in the Methods section (lines 708-710) and as previously described by Zhang et al.

PNAS 2018.

Other issues:

- Page 4. Figure referenced should be Fig. 2E in “This was in contrast to 139 dynamically growing and shrinking WT microtubules displaying EB3 comets only at their 140 growing ends (Fig. 1E,..”

Thanks. Typo corrected.

- The sentence on p9 “20%Q+N microtubules show dimer rise more similar to that...” requires citations

We added a citation.

- Figure S3D and table 2. They calculate the persistence length via a curvature distribution. Their values seem quite low, comparable to taxol treated brain MT. They should give the length distribution of the polymers used. Were they calculated at the same growth rate? The fact that the mutant mix is similar to the WT (actually a little larger). Does this imply that lattice strain is a driving factor in the persistence length measurements?

Growth rates were indeed set to be equal by adjusting the tubulin concentration, as stated. This resulted in the different GTP hydrolysis-deficient microtubules having same lengths after same times of growth. Wild type microtubules can undergo catastrophes and therefore naturally have different lengths.

REVIEWER COMMENTS

Reviewer #1 (Remarks to the Author):

This revised manuscript is improved, particularly with respect to the new Figure 5, additional statistics, and the new Figure S4. I think it's safe to say that the work is noteworthy and should be published in Nature Communications. But I continue to be slightly baffled by the lack of attention to detail in the presentation and clarity of the results. Consider the following examples:

- The microtubule in Fig. 1 is too wide (9 protofilaments visible, suggesting an 18 pf microtubule).
- Fig. 1A, spelling error in the word "straight"
- Fig. 2E, some kymographs are wider than others unnecessarily.
- Fig. 3A and 3C, excessive saturation of kymograph insets
- Fig. 3A, three example arrows in top left, only one example in other images.
- Fig. 4A, there's a random 0 in the left margin.
- Fig. 4B, aspect ratio of plot is skewed and it's 2x taller than necessary.
- Fig. 4C, kymographs are not the same width.
- Fig. 5D, E, why is the plus end at the bottom, which goes against convention and the Fig. 1 schematic?

These glitches are mostly annoying, and good graphics are not a substitute for good data. But at the same time, sloppy graphics might indicate a lack of attention to detail overall, and therefore they distract from the impact of the work. While it's true that scientists are not necessarily graphic artists, it's possible to hire graphic artists and/or solicit feedback from others. Please take the time to present this work properly.

Reviewer #2 (Remarks to the Author):

The authors have addressed my questions. I only have one more comment. In the Discussion (Line 345-352), the authors further interpret the non-linear growth rates of GTP-hydrolysis deficient

mosaic microtubules with the proportion of α E254Q-tubulin in the mixture. The authors think this non-linearity is due to not only the properties of the soluble tubulin dimers but also the conformation of the growing microtubule end itself. The reviewer believes this conclusion is a bit stretched, as the authors did not look into the structures of microtubule ends. The other possible explanation will be the conformational changes when incorporating tubulin subunits into the lattice. It will be more appropriate for the authors to include other alternative explanations.

Reviewer #3 (Remarks to the Author):

In their revision the authors addressed some of the problems they had with lack of quantitation and statistical analysis and transparent methods. Unfortunately, one of the key problems of the manuscript, the one related to their cryo-EM reconstructions, has not been resolved. The authors report reconstructions that are in the 4Å or better resolution range, but fail to show in ANY of their figures fits of their models to the maps or close-ups of the nucleotide binding pocket to show the identity of the nucleotide. No reports are provided from the deposition to the PDB or EMDB (and the deposited datasets are not yet available, so it is impossible to evaluate any of their data in the absence of figures showing quality of the maps and the model fits).

The differences in dimer rise values reported between the different lattices are very small, for example a difference between 82.3 versus 81.5 ie less than 1 Å. This could be due to errors in the reconstructions. It not uncommon in the field to split the dataset into several subsets in order to obtain errors, a suggestion I made in my original review. But the authors state in their response: “We note that such statistical analysis is currently not the standard in the field,..” This is either dishonest or uninformed because several (if not most) studies that report lattice compaction have now taken exactly the approach of splitting the datasets to evaluate the robustness of these values (see publications from the Nogales and Moores labs in the last 5 years, at least).

Minor comments:

1.The emphasis that the EtoQ is only 1Da change in a large protein is disingenuous or overly simplistic. It is changing the charge and the H-bonds for this residue, so it is not just a simple 1Da change out of a 100 Kda protein! EtoQ mutations have been the standard in the GTPase field for several decades!

2. The authors state at the end of their paper that “the mosaic microtubule approach promises to also be valuable in other areas of microtubule biochemistry, for example, for the investigation of the compositional complexity of microtubules with respect to different tubulin isotypes 48 or different post-translational modifications 49.” Can the authors explain what they mean by this ? Do they simply mean the mixing of different types of tubulins, which has been done already for recombinant tubulin of different PTMs or isotypes?

REVIEWER COMMENTS

Reviewer #1 (Remarks to the Author):

This revised manuscript is improved, particularly with respect to the new Figure 5, additional statistics, and the new Figure S4. I think it's safe to say that the work is noteworthy and should be published in Nature Communications. But I continue to be slightly baffled by the lack of attention to detail in the presentation and clarity of the results. Consider the following examples:

Authors' response: We are glad that the reviewer continues to find our results interesting and important and acknowledges the improved presentation of our results. We further improved the presentation as suggested and as described below.

- The microtubule in Fig. 1 is too wide (9 protofilaments visible, suggesting an 18 pf microtubule).

Corrected. 7 instead of 9 protofilaments are now visible.

- Fig. 1A, spelling error in the word "straight"

Typo corrected.

We have also removed the word "straight" from panel C of Fig. 1 to avoid confusion between lattice supertwist and tubulin conformation. Moreover, in panel B, we show now for the expanded lattice also the absolute value for the dimer rise (83.0 – 83.5 Å) instead of the relative change (+ 1.0 – 1.5Å) to improve clarity.

- Fig. 2E, some kymographs are wider than others unnecessarily.

The kymographs have now the same width.

- Fig. 3A and 3C, excessive saturation of kymograph insets

We have adjusted the display of the kymographs to reduce saturation and added a statement to the Methods explaining how kymographs are displayed: "For the kymographs in Figure 3, the maximum contrast value was increased by 25% compared to the individual images (multiplied by 1.25) to allow for better visualization of slight intensity variations." (Lines 575-577)

- Fig. 3A, three example arrows in top left, only one example in other images.

Correct. The arrows point to the microtubules from which the displayed kymographs were generated. We added a statement to the legend explaining this (Line 1077-1078). Because α E254N microtubules show different lattice states, three example kymographs instead of one are shown in this case.

- Fig. 4A, there's a random 0 in the left margin.

Has been removed.

- Fig. 4B, aspect ratio of plot is skewed and it's 2x taller than necessary.

Aspect ratio has been changed.

- Fig. 4C, kymographs are not the same width.

Kymographs of the α E254Q and 20%Q+N microtubules have the same width. Kymographs of WT and α E254N microtubules are less wide because the microtubules explore less space, allowing to place them side by side to keep the figure compact. All kymographs are displayed at the same scale.

- Fig. 5D, E, why is the plus end at the bottom, which goes against convention and the Fig. 1 schematic?

We agree. We changed the orientation, and the plus end is now at the top. Additionally, to better visualize the changes in dimer rise and protofilament twist between the superimposed models, we have reduced the thickness of the ribbons and included both lateral and luminal views for each superimposition.

Reviewer #2 (Remarks to the Author):

The authors have addressed my questions. I only have one more comment. In the Discussion (Line 345-352), the authors further interpret the non-linear growth rates of GTP-hydrolysis deficient mosaic microtubules with the proportion of α E254Q-tubulin in the mixture. The authors think this non-linearity is due to not only the properties of the soluble tubulin dimers but also the conformation of the growing microtubule end itself. The reviewer believes this conclusion is a bit stretched, as the authors did not look into the structures of microtubule ends. The other possible explanation will be the conformational changes when incorporating tubulin subunits into the lattice. It will be more appropriate for the authors to include other alternative explanations.

Authors' response: We thank the reviewer for pointing this out. We have expanded our discussion, as suggested, stating now that the observed non-linearity may also be due to different conformational changes of the incorporating tubulins at the growing ends.

Reviewer #3 (Remarks to the Author):

In their revision the authors addressed some of the problems they had with lack of quantitation and statistical analysis and transparent methods. Unfortunately, one of the key problems of the manuscript, the one related to their cryo-EM reconstructions, has not been resolved. The authors report reconstructions that are in the 4Å or better resolution range, but fail to show in ANY of their figures fits of their models to the maps or close-ups of the nucleotide binding pocket to show the identity of the nucleotide. No reports are provided from the deposition to the PDB or EMDB (and the deposited datasets are not yet available, so it is impossible to evaluate any of their data in the absence of figures showing quality of the maps and the model fits).

Authors' response: As requested, we have sent to the editor all PDB validation reports, maps and models obtained for our four new structures. Furthermore, we have now included a new supplementary figure (Suppl. Fig. 6G) in which we show examples of the quality of the fitting of our models to the symmetrized maps.

The differences in dimer rise values reported between the different lattices are very small, for example a difference between 82.3 versus 81.5 ie less than 1 Å. This could be due to

errors in the reconstructions. It not uncommon in the field to split the dataset into several subsets in order to obtain errors, a suggestion I made in my original review. But the authors state in their response: “We note that such statistical analysis is currently not the standard in the field,..” This is either dishonest or uninformed because several (if not most) studies that report lattice compaction have now taken exactly the approach of splitting the datasets to evaluate the robustness of these values (see publications from the Nogales and Moores labs in the last 5 years, at least).

As suggested by the reviewer, we have split our data sets into three according to the procedure reported in references 20 and 30. The results are now reported in Table 3 and Materials and Methods (Lines 715-719). We found errors in the same ballpark as previously reported in references 20 and 30.

During the splitting and reprocessing of our data, we realized that the dataset of the EB3-decorated 20%Q+N mosaic microtubules did not contain enough particles to provide reliable measurements of the lattice parameters in the smaller, split datasets. To increase the number of particles compared to the previously automatically picked particles by RELION, we manually picked the particles, which indeed resulted in a higher number of 13-prot filament particles. This enhancement in the number of particles improved the quality of our dataset, and we have updated Table 3 and Supplementary Tables 2, 3 and 4 with the new parameters accordingly.

We also identified an error in the model building of undecorated mosaic 20%Q+N microtubules which we corrected, and the model was subsequently refined against its corresponding symmetrized map. Supplementary Table 4 has been updated to reflect these changes.

Minor comments:

1. The emphasis that the EtoQ is only 1Da change in a large protein is disingenuous or overly simplistic. It is changing the charge and the H-bonds for this residue, so it is not just a simple 1Da change out of a 100 Kda protein! EtoQ mutations have been the standard in the GTPase field for several decades!

We have removed this statement, as it does not seem to be helpful.

2. The authors state at the end of their paper that “the mosaic microtubule approach promises to also be valuable in other areas of microtubule biochemistry, for example, for the investigation of the compositional complexity of microtubules with respect to different tubulin isotypes 48 or different post-translational modifications 49.” Can the authors explain what they mean by this? Do they simply mean the mixing of different types of tubulins, which has been done already for recombinant tubulin of different PTMs or isotypes?

Yes, we intend here to close with a more general statement about the usefulness of mixing different tubulin variants, acknowledging at the same time previous work studying other aspects of microtubule structure and behavior (effect of tubulin isotypes and different post-translational modifications).

REVIEWERS' COMMENTS

Reviewer #2 (Remarks to the Author):

The authors have nicely addressed reviewers' questions. I do not have further comments.

Reviewer #3 (Remarks to the Author):

The authors have addressed my comments. In terms of interpretation and correlation between lattice expansion and stability, a thing they might consider is that yeast microtubules do not undergo this compaction between GTP-like and GDP lattices.

Also, regarding citations, the approach of using mixtures of isotypes and PTMs has been done by several groups now and they should consider a more balanced citation practice. Here are a few:

Pamula et al. JCB 2016 (isotypes)

Vemu et al. MboC 2017 (isotypes)

Chen et al. Dev Cell 2021 (PTMs)

Diao et al. JMCB 2021 (isotypes)

Response to the concerns of reviewer 3

Reviewer #3 (Remarks to the Author):

The authors have addressed my comments. In terms of interpretation and correlation between lattice expansion and stability, a thing they might consider is that yeast microtubules do not undergo this compaction between GTP-like and GDP lattices.

Authors' response: Thank you for your comments. We are aware of the structural properties of yeast microtubules. However, the focus of our work is on dynamic instability and the GTP cap of mammalian microtubules. To be clearer about this, we have edited our text and specify now our emphasis on 'mammalian' microtubules more explicitly where appropriate, both in the Introduction and in the Discussion.

Also, regarding citations, the approach of using mixtures of isotypes and PTMs has been done by several groups now and they should consider a more balanced citation practice. Here are a few:

Pamula et al. JCB 2016 (isotypes)

Vemu et al. MboC 2017 (isotypes)

Chen et al. Dev Cell 2021 (PTMs)

Diao et al. JMCB 2021 (isotypes):

We have rephrased our sentence regarding the use of mosaic microtubules and have included these citations as suggested.